# Automation of Life Cycle Assessment—A Critical Review of Developments in the Field of Life Cycle Inventory Analysis

Bianca Köck * , Anton Friedl , Sebastián Serna Loaiza , Walter Wukovits  and Bettina Mihalyi-Schneider

Institute of Chemical Engineering, TU Wien, Getreidemarkt 9, 1060 Vienna, Austria
* Correspondence: bianca.koeck@tuwien.ac.at; Tel.: +43-664-8635967

**Abstract:** The collection of reliable data is an important and time-consuming part of the life cycle inventory (LCI) phase. Automation of individual steps can help to obtain a higher volume of or more realistic data. The aim of this paper is to survey the current state of automation potential in the scientific literature published between 2008 and 2021, with a focus on LCI in the area of process engineering. The results show that automation was most frequently found in the context of process simulation (via interfaces between software), for LCI database usage (e.g., via using ontologies for linking data) and molecular structure models (via machine learning processes such as artificial neural networks), which were also the categories where the highest level of maturity of the models was reached. No further usage could be observed in the areas of automation techniques for exploiting plant data, scientific literature, process calculation, stoichiometry and proxy data. The open science practice of sharing programming codes, software or other newly created resources was only followed in 20% of cases, uncertainty evaluation was only included in 10 out of 30 papers and only 30% of the developed methods were used in further publication, always including at least one of the first authors. For these reasons, we recommend encouraging exchange in the LCA community and in interdisciplinary settings to foster long-term sustainable development of new automation methodologies supporting data generation.

**Keywords:** life cycle assessment; life cycle inventory automation; machine learning; molecular modeling; knowledge engineering; digital twins; process simulation; artificial neural networks; open data

## 1. Introduction

Having evolved over decades, life cycle assessment (LCA) has become one of the most important environmental assessment methods and is widely accepted for evaluating chemical processes and products [1,2]. With new areas, such as green chemistry or the development of biorefineries, to make production less dependent on fossil raw materials, science in this field often deals with techniques in the early stages of development. LCA not only helps with product or process comparisons but also with the identification of hotspots along the different process steps [3].

LCAs consist of four steps based on the ISO14040 and ISO14044: the Goal and Scope phase (e.g., definition of system boundaries), the life cycle inventory analysis (e.g., data collection of input and output data), the life cycle impact assessment (conversion into environmental impacts) and the interpretation (evaluation of results) [4]. Performing an LCA requires a great deal of time and data, both of which are often limiting factors in the development of new processes that are still in laboratory or pilot stages. This paper, therefore, reviews current developments in the automation of life cycle inventory (LCI) creation, as this can help to make the integration of environmental impacts into decisions during the technology development process easier, less time-consuming and can improve data quality.

As the most time- and resource-intensive, the second step in an LCA, the LCI, deals with information and data collection for creating the processes needed for the LCA study [5,6]. Information collected for the product system typically covers a considerable volume of data [7], including the flows to and from processes. This involves elementary flows (resources and emissions but also land use), product flows (goods and services as products in a process) or waste flows, as well as statistical data for, e.g., the market mix or process [4]. LCA covers activities which take place at different times and places [8]. Additionally, data quality and uncertainty distribution should also be collected for the parameters [5]. In a typical LCA, thousands of unit processes are usually used [9].

The product system, which, in many cases, involves mapping a complex globalized network consisting of thousands of interlinked human activities [10,11], is made from decisions taken by different individual economic actors, based on economic, legal and environmental indicators, and linked in the LCA based on the functional unit [8]. With regard to the influence of the decision maker, for which the LCA is carried out, the system is divided into fore- and background systems [8], which are often defined as follows:

Foreground system: The foreground system consists of processes which are under the control of the decision maker for which an LCA is carried out. They are called foreground processes.

Background system: The background system consists of processes on which no, or, at best, indirect influence may be exercised by the decision maker for which an LCA is carried out [8,12].

As shown by these definitions, depending on the goal, the author and client, there can be different fore- and background systems and methods, and the possibilities for data procurement can vary.

Another definition of the two systems can be that the foreground system is the one where data collection efforts are conducted, reflecting the space of action [9,13], while the background system models the remaining activities through the use of generic data from LCI databases [9] or public and official national statistics, company websites or previous research published in journal papers [5]. Usually, datasets in the background system cover 99% of unit processes, while, in rare cases, the foreground system exceeds 5% of all processes in the system [9].

The following review will analyze methods with automation potential for data collection for both fore- and background systems. It is emphasized that the boundaries of the two are floating, and, also, depending on time and budget resources, there can be quality in foreground data.

Categories are developed or identified based on the objectives behind the method as well as the data source. Using empirical and bibliographic analysis, possible trends over time and further developments in the methods are identified. In the Discussion and Conclusions, recommendations will be extracted from the analysis for the applications in chemical and process engineering, especially biorefineries, which deal with processes and products in low technology readiness levels (TRLs).

## 2. Materials and Methods

### 2.1. Aims, Goals and Research Questions

For the analysis, the following research questions were defined:

Which methods with automation potential for data generation for the LCI phase can be found in the literature, focusing on the past 15 years. What methods are used and for what purpose is automation used in the LCI?

Are there examples of LCA available in journal papers, where the automation technique was used after development? With which frequency were the proposed methods used and what are their advantages and disadvantages?

How transparent is the study—is, for example, Supplementary Material with comprehensive information available? Can the developed automation techniques be easily transferred or used for further LCAs, where, for example, the code or software is available online?

If statistical models or methods of artificial intelligence are used, how and with what volume of data were they trained, were their advantages or disadvantages mentioned and are there similarities between the different LCAs? Is the model code used open access?

*2.2. Scope*

Our study assesses the automation efforts and methods already implemented in the LCI phase, focusing on the last 15 years. Automation, which, in this context, means the complete or partial replacement of manual human work steps or work by machine, can serve different objectives: reductions in epistemic uncertainty sources through the possibility of collecting more data at the same time, the acceleration of the implementation of individual but similar LCAs due to automation of the data input or the connection between fore- and background data.

The following focuses, in particular, on the LCI, which includes data collection, data calculation and allocation of flows and releases (ISO 14040: 2006 + Amd 1:2020 [14]). Data collection includes energy, raw material, ancillary and physical inputs, as well as products, co-products and waste. "Emissions to air, discharge to water and soil and other environmental aspects" must also be gathered. In the data calculation step, the data are validated and related to unit processes, reference flows and the functional unit to calculate the results of the inventory step. Allocation procedures can be used for systems with multiple products or recycling systems, as shown in Figure 1 [15,16].

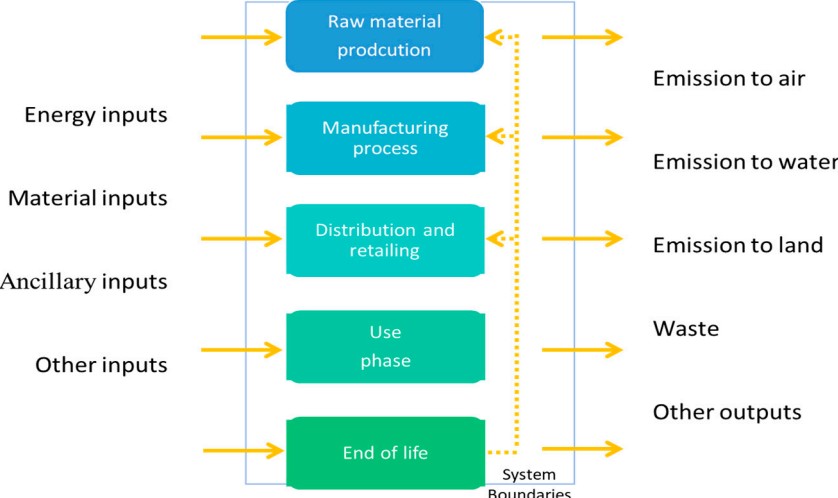

**Figure 1.** Life cycle inventory phase (modified from Bachmann, Hidalgo and Bricout [15]) and DIN Deutsches Institut für Normung [16].

*2.3. Methods and Material*

The literature was sifted systematically and we included only full-text available primary literature from journal or conference papers written in English, which either perform or give guidelines for parts of the LCI phase.

After screening the papers, especially with regard to the fulfilment of the criterion of automation (potential) in the LCI phase and quality of description of the methodology, 30 papers and related articles (based on similar methods by the same authors) were analyzed in more detail in accordance with our research questions.

The criterion of automation was fulfilled if typical human work steps in the LCI phase of the LCAs were completely or partially replaced by machine work, such as transferring and searching for data or filtering out relevant in- and output flows.

Authors of the identified literature were also scanned for further publications in this area to identify which development or model was further used. The literature research was conducted based on LCAs or method description papers for LCA dealing with the described parts of an LCI, initially covering the last 10 years (2012–2022) and, subsequently, since some interesting models were found earlier (2008/2009), extended to 15 years.

## 2.4. Delimitations

The focus of this paper is on the derivation of automation potential for the field of chemistry and process engineering, especially for biorefinery development. For this reason, the literature sources evaluated mainly include general methods developed without specific focus or LCAs from this area, and no literature from areas, such as construction, electrical engineering, services, etc., was considered. The conditions and requirements for the respective LCAs can vary greatly from one field to the next and the search was, therefore, confined to the mentioned fields.

However, automation in the field of inventory assessment and methods such as machine learning are mostly generally applicable to a wide range of LCAs, and the relevant parts of this paper may also add value to related research fields.

As this study is focused on LCAs, no other environmental assessment was considered for the selection of papers.

## 2.5. Classification

In order to categorize similar approaches, existing categories for LCI creation were considered. Although there are several classification schemes for LCI data generation, including differentiating bottom-up, top-down and hybrid approaches [17] or computational approaches, economic input–output analysis and the combinations of those two approaches [18], we decided to use the more detailed hierarchy created by Parvatker and Eckelmann [19] on the topic of "Chemical Life Cycle Inventory Generation Methods", which was expanded by Giesen et al. [20] with regard to ex ante claims. We aimed to evaluate whether the automation in LCI generation can also be seen in LCA literature of the last 15 years with regard to their hierarchy levels. Paravatker and Eckelmann [19] tested their analysis of accuracy and data/time requirements between the different methods on creating LCI data for chemicals on case studies on ABS and styrene, where very large margins of error between the methods, exceeding 500% in some cases, could be observed. As the data and time requirements, especially with low TRL, are often a limiting factor for the LCA, literature on automation techniques was sifted to demonstrate that more accurate methods can, in the future, be used with less time consumption and generate more reliable data. In this research, as process engineers, we focus more on the TRL levels, although the original work on the inverted pyramid (Figure 2) focuses on manufacturing readiness level (MRL). The MRL scale is comparable to the TRL scale [21].

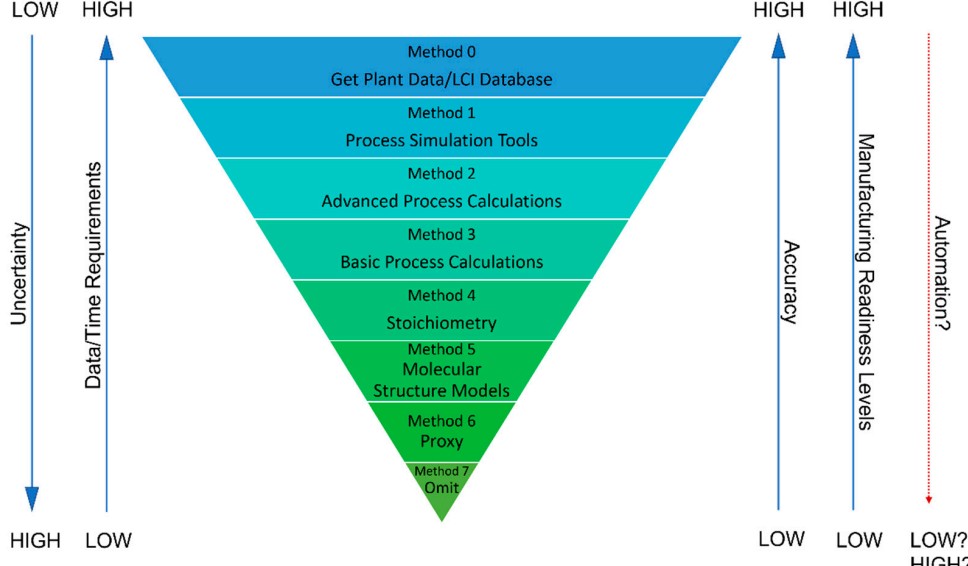

**Figure 2.** For evaluation of LCI data generation, automation methods used the hierarchy adapted with permission from Ref. [19]. 2019, American Chemical Society and Ref. [20]. 2020, Elsevier. MRL = Manufacturing readiness level.

The categories for our analysis are based on the hierarchy elements in Figure 2 and described in more detail in the Results.

### 2.6. Findings from the Papers and Proceedings

With the goal of communicating the methodological approaches, objectives and results of the articles, a summary of the articles is categorized based on the ways in which LCI data can be generated. Special focus is placed on the description of the method, the automation and possible advantages and disadvantages (such as the inclusion of uncertainties).

### 2.7. Bibliometric Analysis

For statistical analysis, keyword identification was performed using the software VOSViewer [22], and the most common categories, years of publication and journals published were identified using Web of Science (analytic tools and citation report) [23] and Microsoft Excel. The authors' relationships with one another were also examined using Research Rabbit [24].

### 2.8. Empirical Analysis

Additionally, an empirical analysis was conducted to research the availability of open science aspects (such as open-source publication, open-source code or software, etc.), as well as an analysis of uncertainty and the used artificial intelligence techniques such as machine learning.

### 3. Results

Building on the classification of Parvatker and Eckelmann [19], presented in the Methods section, categories were merged based on findings in the literature (basic and advanced process calculation). The categories were separated based on the breadth of methods and subject areas (plant data and LCI database), more detailed sub-chapters were introduced (process simulation) and a new category was introduced, namely, data from the scientific literature, as shown in Figure 3.

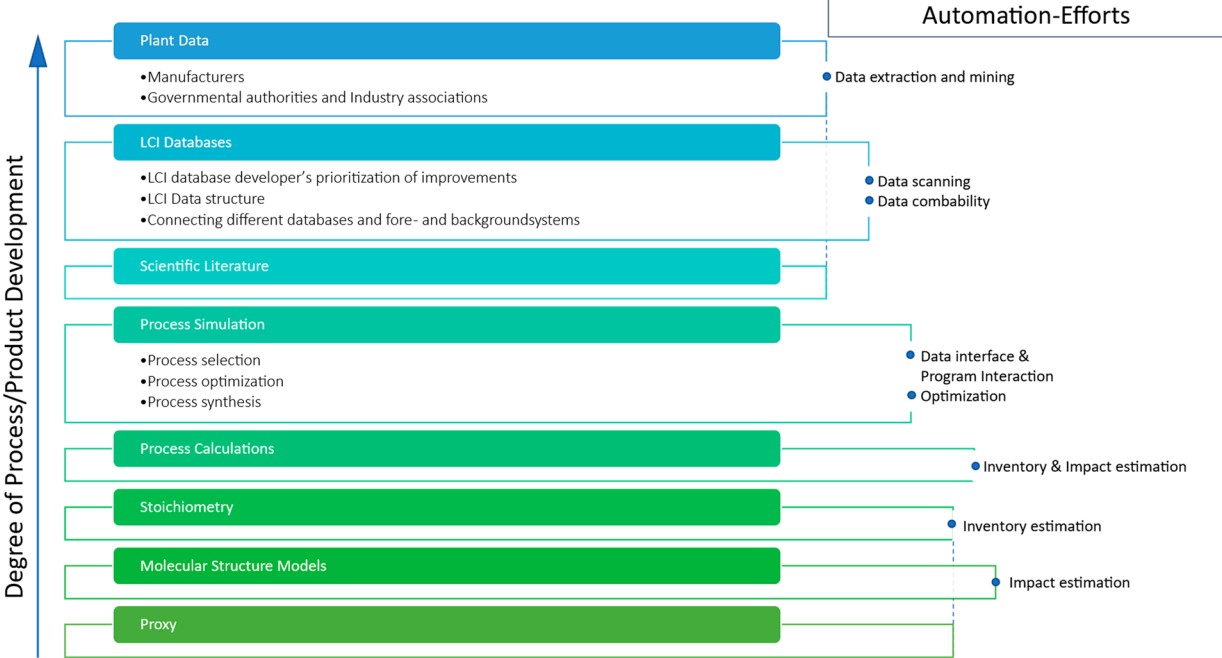

**Figure 3.** Methods of LCI data generation with automation potential identified in the literature.

The hypothesis of a tendency to automation based on the Parvatker and Eckelmann pyramid [19] could not be substantiated, as methods for optimization could be found in

each of these categories. What is striking here, however, is the choice of methods depending on the approach.

*3.1. Plant Data*

Obtaining data directly from the production facility in the right geographical location increases the accuracy of LCA results [25] but is often not possible due to secrecy or poor monitoring. Even when data are received, it is essential to compare process conditions and check for measurement and reporting errors [19].

Plant data can also be gathered from governmental authorities, as many industries report LCA-relevant data such as capacities to them. This, however, extends the scope of the LCA to a larger geographical location and temporal frame [19]. Data can, for example, be obtained from the United States Environmental Protection Agency (EPA) or the European Life Cycle Database (ELCD), which are both integrated in the Global LCA Data Access (GLAD) [26,27]. Additionally, industry associations, such as the International Council of Chemical Associations (ICCA) or the American Chemistry Council (ACC), provide data (Parvatker and Eckelman 2019), and some of these, such as Plastics Europe or the European Solvents Industry Group, also provide them via GLAD [26]. Another source of data is LCAs conducted by companies themselves, either published on their own or in environmental product declarations (EPDs), which communicate life cycle impacts of products according to the product category rules (PCRs). However, these often only contain aggregated results and no detailed process steps nor complete LCI datasets [19]. Another data source, e.g., for toxicity, is Comptox [28], a database from the EPA, which links different data, e.g., from pubmed, Wikipedia, the NIST Chemistry WebBook and Usetox [29,30].

The automation potential in relation to government agencies and industry associations can be observed as follows: Using linked opened data from the EPA [31] on chemicals manufacturing in a standardized way with the outlook on automating the discovery, and use of publicly available data was, therefore, added in the review. The mined data are reported by companies, but a lack of transparency regarding underlying process technologies should be taken into account, as well as possible variation in reporting requirements in the data submission, which can influence the LCI generated [31,32]. However, a comparison with existing LCI data and a case study showed that the information provided is as or more reliable in the areas of air, water, hazardous waste, on-site energy usage and production volumes. Key data gaps were identified, including material inputs, water usage, purchased electricity and transportation requirements [31].

By using linked open data (LOD), the varying degree of uncertainty and heterogeneity of the mined data should be reduced and lead to higher-quality data for the LCI. LOD is a machine-readable format to support facilitated search and retrieval, guided by the emergence of the semantic web, which is an approach to structuring and describing data to enable sharing across multiple systems and applications [31]. For example, LODs from the EU can be found here: https://data.europa.eu/ (accessed on 30 December 2022).

Cashman et al.'s method [31] was also used by Meyer et al. [33] but extended by additional filtering based on metadata to exclude on-site activities not associated with the desired chemical manufacturing. Their study included a comparison of top-down (datamining), bottom-up (simulation) and a statistical approach, and it concluded that the top-down-approach was limited by only being applicable for existing chemicals with reporting requirements of release data and that there were differences in data quality, depending on the generation of the data through measurements (ideal case), estimates or engineering judgment [33]. Other aspects of Meyer et al.'s work are mentioned in the sections below [33] and more recent work, like a method on data mining workflows for modeling chemical manufacturing [34].

With respect to data gleaned directly from a factory's manufacturing process, a study using shortcut models for data extraction in standard operating procedures demonstrated an automation opportunity. Energy use in the chemical industry is often an important factor for environmental analysis, although it is often not very well documented nor monitored

when dealing with batch processes due to their complexity and dynamic nature [35,36]. Pereira et al. [36] proposed a bottom-up shortcut model using the standard operating procedures (SOPs) of factories, where, through a systematic methodology building on basic equations, energy data were extracted and data gaps were filled with default values or expert knowledge. Uncertainties in this approach can be based on the quality of standard process documentation as well as differences to the reality of batch-to-batch variability. The use of default values, model simplifications and assumptions also adds to the uncertainty. In the two case studies examined, the uncertainty was incorporated through fuzzy intervals and the validation results showed "generally a good agreement between reference and predicted values and a good capability of the uncertainty intervals to capture the batch-to-batch variability of steam consumption" [36]. Data obtained from the SOPs using the methods described were used by C. Pereira et al. (2018) [37] in a statistical model based on reaction synthesis types, further described in the section on Section 3.9 in the article.

This method offers little possibility of automation so long as it is not coupled with text analysis algorithms and support schemes for the calculation. However, it can save time for LCI database generators or practitioners compared to basic facility-level data collection. Incorporating the uncertainty in the data quality and using default data are also necessary to secure the findings used with these data. One limitation, however, is the need to access companies' SOPs and ideally to be able to check their accuracy.

Developments in the field of LCA integration in dynamic manufacturing support tools are also ongoing and are summarized in the book *Resource Efficiency in Manufacturing Value Chains* by Blume [38].

### 3.2. LCA Databases

Using LCA databases reduces time, effort and resources for data collection and is often preferred for creating the background system [5,11]. However, the unit processes in the databases are subject to uncertainties, as the required data for them may, for example, be unavailable, incorrect or unreliable [11,39,40], as well as representing only an average condition of a whole country, a given time period as well as different instances of real processes [41,42]. This then makes post hoc improvements necessary, which is evident in the number of database updates [43,44].

Very common LCA databases have thousands of unit processes, such as GEMIS (~10,000 processes [45]), GaBi (~14,000 datasets excl. 1000 complete models [46]) and Ecoinvent (~18,000 processes [47]), which perform systematic updating and improvements in the stored data [11]. Reinhard et al. [41] stated that, to the best of their knowledge, there is no systematically guided way to implement LCI database improvement tasks. Data quality and quantity improvements are more influenced by the external supply of data as well as situation-driven requirements [11]. Therefore, a semi-automated prioritization method was developed and applied on the Ecoinvent 3 database [9] using MATLAB, establishing that across 19 LCIA indicators, 300 datasets were causing 60% of the environmental impacts, mostly in the areas of electricity generation, waste treatment activities and energy carrier provision (petroleum and hard coal) [41]. This method automates the prioritization for updating through systematic selection of processes. However, the uncertainty in the existing processes is not included, although an improvement in data quality is also needed for uncertain processes. Additionally, each product/product system is given equal weight, leading to many similar and connected processes being considered relevant [41].

This method was extended by a causer perspective, prioritizing "processes with exchanges of resources and emissions that are consistently important across product systems and LCIA indicators", and the connector perspective, where "sensitive hubs whose modification can alter the results for the overall database considerably" are prioritized as well [11]. For climate change, three datasets caused 11% of the impacts [11].

Another project dealing with data generation, validation and management decision is the Big Open Network for Sustainability Assessment Information (BONSAI) project, with the main goal of maintaining an open database for LCA data and an open-source toolchain

for the support of LCA calculations to make LCAs more transparent and reproducible [48]. One of their strategies is automation through "scrapers, data mining & advanced algorithms for disaggregation, reconciliation and extrapolation" [49]. The BONSAI board consists of many LCA scientists and LCA software creators who will be mentioned with respect to other topics. These include Chris Mutel (scientist at the Paul Scherrer Institute, principal author of the Brightway2 software framework), Bo Weidema (Professor at Aalborg University, developer of the original SPOLD LCI dataformat and former Executive Manager and Chief Scientist of the Ecoinvent database) and Andreas Ciroth (founder and director of GreenDelta, which developed the LCA software openLCA) [50]. Developments so far can be accessed via Github [48].

Other automation approaches were found regarding changing the LCI data structure. Different data formats bring limitations, and there are approaches to solving the issue of data incompatibility, as the non-standardized LCA data structure inhibits automation efforts.

Building on this problem, new visions for data structure and harmonization have been developed, such as the new data architecture by Ingwersen et al. [51], which combines, stores and annotates LCA data in a resource description framework. One advantage is "making data from diverse sources relatable without the need for a centralized database schema" [51]. The code for the developed harmonization tool can be accessed on Github [52]. This was a basis for the Federal LCA Commons [53], as discussed in a report by Edelen et al. [54]. The data are also managed by the openLCA Collaboration Server [53,55].

Mittal et al. [56] developed a coupled semantic data methodology for automated inventory modeling for chemicals, through splitting inventory modeling data into two concepts with different data needs: "lineage", such as the synthesis pathway, using qualitative data, as well as "process", focusing on the process condition, which is quantitative. These linked open datasets are coupled, for example, by chemical reaction and reaction participants. By using the ontology query language, SPARQL (SPARQL Protocol and RDF Query Language [57]), queries can be deployed to obtain an automated inventory of possible fitting chemicals for concepts of ancestors or parents. Tested on a case study of nylon-6 production, the proposed ontology guides inventory modeling using top-down or bottom-up approaches [56]. Both main authors of the previously discussed papers, Ingwersen and Mittal, were co-authors on the paper by Meyer et al. [33], which used classification trees.

Steubing et al. [58] proposed creating superstructures for future scenario LCI data in the background system, containing different years and all individual exchanges of one specific dataset, instead of creating many individual processes to reduce time and possible mistakes when reconnecting the foreground system by hand to the various background datasets and automatizing the scenario analysis. They illustrated their approach in the open-source software Activity Browser [59], which is built on Brightway2 [60]. In the outlook, they also mention the possibility of applying this approach to foreground systems. Limiting factors in the development so far are the need for standardized future scenarios, the creation and access to a future scenario database and the guidance for the LCA practitioners [58].

Automation potential was also found regarding connecting different databases and fore- and background systems. Automation when using the "Ecoinvent LCI database with scenario data from various data sources" was, for example, achieved by Mutel in his freely available application "Wurst" [61], using Python and the Python-based LCA software Brightway [60].

In some LCA studies, it is necessary to use data from different databases, as appropriate background data are not available in a single database. This can be challenging, however, as it is necessary to find the appropriate data, structure them for use in an LCA as a network of processes and link the processes together to build a product system model (PSM) [62]. To obtain data from the GaBi database or from GEMIS, the related software is required, as it is not freely available for different LCA software. There are also different data formats for LCA data available, including more universal ones, such as EcoSpold, International Reference Life Cycle Data System (ILCD) and JavaScript Object Notation (JSON),

as well as software/database-specific ones, such as Zolca (for OpenLCA), SimaPro CSV (for SimaPro) or GBX (for GaBi). To convert EcoSpold1, EcoSpold2, ILCD and JSON-LD in any combination, the open-source project Global LCA Data Access (GLAD) [26] can be used as well as a tool by GreenDelta, which converts EcoSpold and ILCD [63]. To discover appropriate data, there are portals available, such as the aforementioned GLAD [26], the LCA Nexus (GaBi data no longer available) [64] or the Federal LCA Commons [53]. These search platforms are, however, still in their early stages [62]. To establish a standard for LCA data, the United Nations' Environmental Program "Shonan Guidance Principles" [65] was developed as guidance principles for LCA databases.

Although finding the appropriate data can be challenging, another great challenge right now is linking LCI data from different sources together in a structure that can be reviewed, revised and reused by others [62,66]. An interim solution for doing so without altering the newly created foreground process, which should be linked to different databases with different process designation, is to use bridge processes, which combine, as input, the different possible processes to be connected with the foreground process and give, as output, exactly the input described in the process. The advantages and limitations of this approach are discussed in [67]. Based on this publication, the prototype autoprox, written in Kotlin, is available, which directly communicates with openLCA and the implemented database, generating bridge processes [62,68,69].

Data quality and missing data were evaluated from case to case, and the extent to which the processes to be linked fit together is subject here to the information and decisions of the practitioner.

Another recent study, by Kuczenski et al., addressed the challenge of connecting primary collected data for the foreground system with background data from databases in an automatic way, by having three LCA software developers work on independent software prototypes. Their task was to convert a Bill of Materials (BOM) into a foreground model and then link it with background datasets [62]. Three initial prototypes were tested and can be found on Github. The first prototype is pslink [70] and works unsupervised with OpenLCA over the interface olca-py [69] It is the only model which is linked to Ecoinvent. The second model is called antelope [71] and works supervised with the LCA-software Antelope LCA, while the third model perdu [72] produces data readable by Brightway and OpenLCA, and needs supervision by the matching of datasets.

These prototypes already facilitate automated install or embedding, and they can interact with common used LCA software, for example, openLCA [73] and Brightway [60]. Additionally, they use a standard exchange format such as JSON-LD. With greater investment of time and integration in LCA software, the tools would be more accessible for LCA practitioners. So far, none of their models have been able to transform the product specification into a complete product system model without manual intervention. Although the preselected linking options from the prototypes can contribute to better accuracy, the practitioner still needs to select the processes carefully with respect to data quality and possible difference in flow or process compositions. Another approach to connecting back- and foreground data is ontology design patterns, which could also be enhanced to address the mentioned problems in more detail when in conjunction with similarity matching.

A previous approach to automated linkage from different data sources based on ontologies was also developed by Kuczenski et al. [74]. Based on a joint workshop of ontology engineers and LCA practitioners in 2015, a model called ontology design patterns (ODPs) was created to derive a description of an LCA "catalog" that can be used to express the semantic content of a data resource. This should reduce the problems of heterogeneous formats of different databases and goes beyond widely used exchange formats, such as Ecospold and ILCD, as it not only addresses the syntactic interoperability but supports the interpreting of different data and the evaluation of the datasets' appropriateness for the application. Interoperability and interpretability problems could be addressed via automation by changing different labels between databases (e.g., for Designations of Regions, such as the EU or unit conversion). The semantic model contains three classes

or entity types: "Activities", "Flows" and "Flow Quantities" as well. There were also two relationships: "Exchange" is an established relationship between an activity instance and a flow instance and "characterization" is an established relationship between a flow and a flow quantity. The format of the generated catalogs are text documents in JSON [74].

One strength is, for example, that these catalogs can be used for advanced queries, as the used search terms can be found in different data description positions depending on the database used. Therefore, searching one keyword in two different databases, such as Ecoinvent and GaBi, cannot appropriately cover the existing data right now. Using semantic modeling tags, which appear frequently together with a given search term and could be extracted automatically, along with codes to reproduce the search results, could be provided with the catalogs. Additionally, similar or equivalent entities can be identified by using text descriptions that have similar semantic content. Automation can be accomplished here through advancing the search by metrics that yield numerical scores based on the actual words in the process name, such as the Jaccard Index, used as an example in the publication and locating synonyms in the queries by using online resources such as WordNet [75] or machine learning algorithms such as Word2Vec [74,76]. The Python-based repository for storing catalogs of LCI data is available on Github [77].

Implemented data conversion and importing are integrated in the open-source process simulation software BioSTEAM [78], which has an implemented LCA module based on Brightway2. This setup helps to overcome challenges when connecting foreground data from the simulation with LCA databases [79].

### 3.3. Scientific Literature

There are many scientific articles from which data can be extracted and used for LCAs. However, this category was not inserted in the pyramid of LCI data sources in a publication by Parvatker et al. [19], and, depending on the specific data extracted from the paper (LCI data, LCIA data, experimental data, which have to be upscaled to LCI data use . . . ), it can be inserted in different levels of the pyramid. Though they all have this in common, the textual format and heterogeneous structure create obstacles when extracting them [80].

Lousteau-Cazalet et al. [80] presented a decision support system (DSS) for eco-efficient biorefinery process selection using an ontology-based semantic approach, which links open data and offers automated reasoning. The DDS used knowledge engineering (KE) based on semantic web technologies, soft computing techniques (fuzzy logic and belief theory) and environmental factor computation, aiming to "structure experimental information and express it in a standardized vocabulary", as well as assessing the data source reliability and computed and visualized green indicators [80,81]. For data management, extraction and semantic annotation of data, as well as data source reliability assessment and bipolar flexible querying, @Web, an online tool funded by the French National Institute for Agricultural Research [82] was used. A case study comparing different pre-treatment processes for sugar yield after enzymatic hydrolysis was undertaken, though there was no strategy to deal with missing data needed to calculate the implemented environmental indicators (for example, missing energy data) [80].

Belaud, co-author of the article mentioned above, further developed KE models for the semi-automated integration of literature sources with other scientists over the years and, in 2021, presented a pipeline based on the LCA steps in publications with Prioux et al. [83–85]. Their pipeline extends the LCA phases with step 2 "Data architecture", where KE methods extract information from scientific articles and expand the last phase ("Interpretation") with visualization and analysis via machine learning methods and clusters [83]. For the data extraction in the article from Belaud et al. [83], the previously mentioned French National Institute for Agricultural Research [82] @Web platform was used, as well as an internal software based on Microsoft Excel [83]. The extracted data were then written in standardized LCA data formats and connected with the Ecoinvent database v3. Data validation was accomplished using the pedigree matrix method [86], and mass balance was verified manually by process engineers [83] or with the commercial simulator ProSim+ [84].

The LCIA was calculated using SimaPro and the ReCiPe 2008 or 2016 method [83,84]. For the visualization of the impact in step five, classical multidimensional scaling (MDS) and a clustering algorithm (k-means) were used [84].

The case studies performed deal with the pre-treatment of rice straw and corn stover in the context of a lignocellulosic biorefinery [83,84]. As is often the case in this area of research, the data are from experiments in the lab, which means they have low technology readiness levels (TRLs) (1/2). No scaling to higher TRLs has been included in the model to date. Another limiting factor is the manual effort required by mass calculations in process simulation software or by process engineers to check the data quality and calculate missing data. By using data source reliability assessment beforehand, a pre-evaluation of the data quality is included [84].

Another publication worth mentioning (not included in the evaluation as published after the time horizon and focusing on one impact category and not the LCI) is by Shavalieva et al. [87], who trained a predictive model on the aquatic toxicity using semi-automated knowledge extraction of scientific literature.

*3.4. Process Simulation*

Developing a simulation of a complex industrial process is a growing field, including keywords, such as "digital twins" or "smart manufacturing". Its potential from a technical point of view is that it can also create data and reduce uncertainty for LCA models [88].

Parvatker and Eckelman [19], who compared different data sources, showed in their case study on ABS and styrene that for greenhouse gas potential (GWP), the results are similar to the primary data from plants, but for most other impact categories, the process simulation did not perform as well as expected (out of 18 midpoint indicators from the impact assessment method ReCiPe, only 4 were within a 10% range and another 4 within a 25% range). The authors recommend limiting LCI data from process simulation to GWP and cumulative energy demand (CED) impact calculations. Disadvantages also include the expertise required and the duration [19]. However, process simulations are quite common in process development or optimization, so they can be a valuable source of data for the parallel or retrospective LCA.

In a comparison of top-down (data mining), bottom-up (simulation) and statistical modeling approaches, Meyer et al. [33] modelled their case study of cumene manufacturers using an exciting flowsheet by ChemSep19 in the free-of-charge CAPE-OPEN to CAPE-OPEN (COCO) simulation environment. They concluded that the simulation quality depends on knowing the manufacturing process to make it representative of real-world operations, and scored the data quality for their case study compared to the other methods the lowest, as the simulation data used were over a decade old. However, the resource and time requirements were low based on the freely available computational tools and plans. Additionally, they also stated that the required knowledge and training for the simulation are high. In their purpose-driven reconciliation framework, they, therefore, stress the need to choose the data mining method based on the aim of the research, for example, not creating high-quality release estimates by process simulation when screening is the only intended assessment [33].

There is a risk of overlooking important environmental factors by conducting less rigorous estimations when designing a new process, because catalysts, reaction promotors, solvents or other material and machinery used can affect the sustainability of different process opportunities [32]. As experimental data can be rare in the first TRLs, the development of new processes is often supported by process simulation, which provides the opportunity to explore different scenarios despite the lack of existing data. However, process simulation is also used for optimization in large-scale implemented processes at high TRLs and, in some cases, it is used specifically to generate data for LCAs (e.g., [89–91]).

To date, chemical process simulations mostly focus on techno-economic considerations, and environmental issues are often addressed in a later step for process selection (as in [32,92]). This is layer one in Kleinekorte et al. [93], who defined three areas of integration

of LCA in process design. The second layer is integrating the LCA in a feedback loop on a fixed-flowsheet for process optimization, and the third layer even integrates it from the outset into the flowsheet generation via process synthesis by, e.g., "selecting options from a superstructure or by altering the design via evolutionary algorithms" [93]. A visual overview of the three types can be seen in Figure 4.

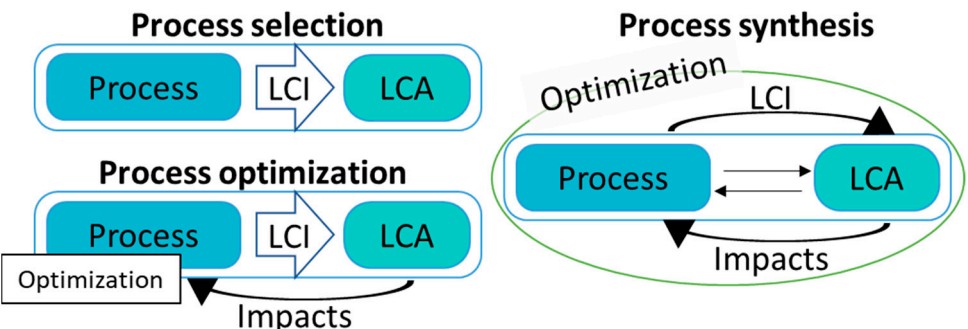

**Figure 4.** Integration of life cycle assessment (LCA) and process design based on research byK-leinekorte [93]. Process selection: evaluation of fixed process structures by LCA. Process optimization: integrated feedback loops of environmental impact on process design in process optimization. Process Synthesis by using algorithms. LCI = Life Cycle Inventory.

Although these layers are defined focusing on the integration of LCA and process simulation with the purpose of choosing or designing more environmentally friendly processes, the automation aspect for obtaining LCI data varies greatly between these three steps, as two (or more) software interact more in each further step. All layers could benefit from automation via interfaces between the process simulation and the LCA software to make the data transfer more efficient. Better cooperation between these two areas could increase the data quality and completeness of LCA [32].

*3.5. Process Selection*

Process selection refers to the mere use of LCAs for choosing a process after simulating, without any feedback on the process design development itself (used, for example, in [94–97]). It is also used in terms of hot-spot analysis to identify parts of the processes with the highest environmental impact (e.g., [93,98,99]). The degree of automation visible in the literature in this layer was very low. There were, however, a few successful implementations, such as the process sustainability prediction (PSP) framework with CAPE OPEN [100,101], tested with frequently used process simulation software (PROII, Aspen Plus, COCO/COFE). The Visual Basic-based framework then uses data obtained by the toxicological database and molecular modeling to calculate environmental impacts with the WAR algorithm [102] and support process selection. Unfortunately, a screening of the citations did not show any reuse of their methodology, but Morales-Mendoza et al. [103] based their work on it [103]. A comment by the Co-lan website administrators [101] recommends CAPE-OPEN flowsheet monitoring as an easier approach, developed after the PSP framework was published. The framework is available upon request [100]. Although there is much potential for making the interaction of process simulation and environmental calculation smoother, in the applications mentioned above, a small range of environmental impact indicators was calculated, limited by the toxicological data available.

Another algorithm used to estimate environmental impact directly from the process simulation flowsheet is the potential environmental impact (PEI) algorithm by Vincent et al. [104], used in the process selection by Mio et al. [105]. However, another use of this framework could not be found, even in the lead author's later publications. A co-author of this paper was Maurizio Fermeglia, whose framework was mentioned above.

Morales-Mendoza et al. [103,106] developed an automated coupling framework, the Ecodesign Framework, which was tested on the process simulation software COCO, ProSim Plus and Aspen HYSYS. The energy production modeling and emission computation were

executed in the energy production simulator Ariane ProSim SA. The LCIA calculation was conducted in SimaPro. The authors highlighted the limitations in the common LCA software interfaces for using embedded LCA models. The integrated data enhance the LCI sensitivity analysis [88,103]. However, in future work, uncertainties should be incorporated by confidence intervals and using Monte Carlo analysis of fuzzy concepts [103]. The ecodesign approach was later extended by a multicriteria decision-making methodological framework [107] and Azzaro-Pantel [108] expanded the developed framework for food manufacturing.

A Microsoft Excel and Visual Basic-based approach called SimLCA was developed by Gillani [109,110], combining SimaPro similar calculation procedures with data from the process simulation ProSim Plus. It was based on a three-step structure (system, business process and agrochemical process) and the involvement of five types of decision makers. Through coupling with the process simulation, automatic updating of LCI and LCIA data was achieved. Uncertainty was incorporated using the pedigree matrix [109].

The process–product–enterprise (P$^2$E) approach [111,112] from two of the authors who worked on the previously mentioned SimLCA (Belaud and Montréjaud-Vignoles) enables process simulation interaction for process comparison and hot-spot analysis for food production, with a case study on olive oil production. Their tool focused on the evaluation of the process–product–enterprise system, as opposed to laboratory data, and the integration approach can be seen in Figure 5. The approach involves converting data directly from process simulation to a product model and sustainability assessment. They also included social aspects (collected qualitatively and semi-quantitatively) as well as multi-criteria analysis. While using Microsoft Excel and SimaPro for the inventory and impact calculations in their first publication in 2014, they then proposed an in-house VBA-application called "&cOlive" for the calculation, including a sheet for the process simulation, which does not yet include energy calculations. Unfortunately, the framework could not be accessed on the internet.

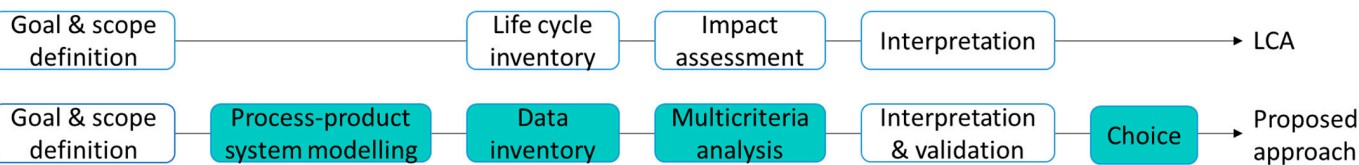

**Figure 5.** Process–product–enterprise (P$^2$E) approach structure modified from [113].

The thermo-environomic optimization methodology by Celebi et al. [113] was adopted from a framework by Gassner and Maréchal [114] and aims to make comparisons of different pathways for biorefineries, ranking them according to economic performance, environmental impact and energy requirement. The model is visualized in Figure 6. The ranking is established "according to the objective function with integer cuts constraints (ICC)" methodology proposed by Maronese et al. [115]. The simulated parameters for their case studies on wood to sugars (biochemical conversion) and wood to syngas (thermochemical conversion) were obtained in Aspen Plus [116] and Belsim VALI [117] and combined in a hybrid superstructure, containing "feedstock and raw materials units (Resources), process conversion technology, units (Units), products units (Services) and layers which correspond to the mass and energy balances nodes in the system". GWP100 was then calculated with formulas, including material and energy flow data (such as fuel and electricity) form Ecoinvent, though it is not mentioned whether automated linkage or selection was used. Defining an MILP problem for the economic and environmental performance, 150 solutions were calculated in the Lua-OSMOSE framework, a platform hosted by the École Polytechnique Fédérale de Lausanne in Switzerland, which allows for linking several types of flowsheeting software and provides analysis tools (optimization, sensitivity analysis, etc.). The OSMOSE framework is available on request [118], but some pages (such as Wiki) were not working. More on the integration from LCA into the OSMOSE framework can be found in Yoo et al. [119], Granacher et al. and Kermani [120,121].

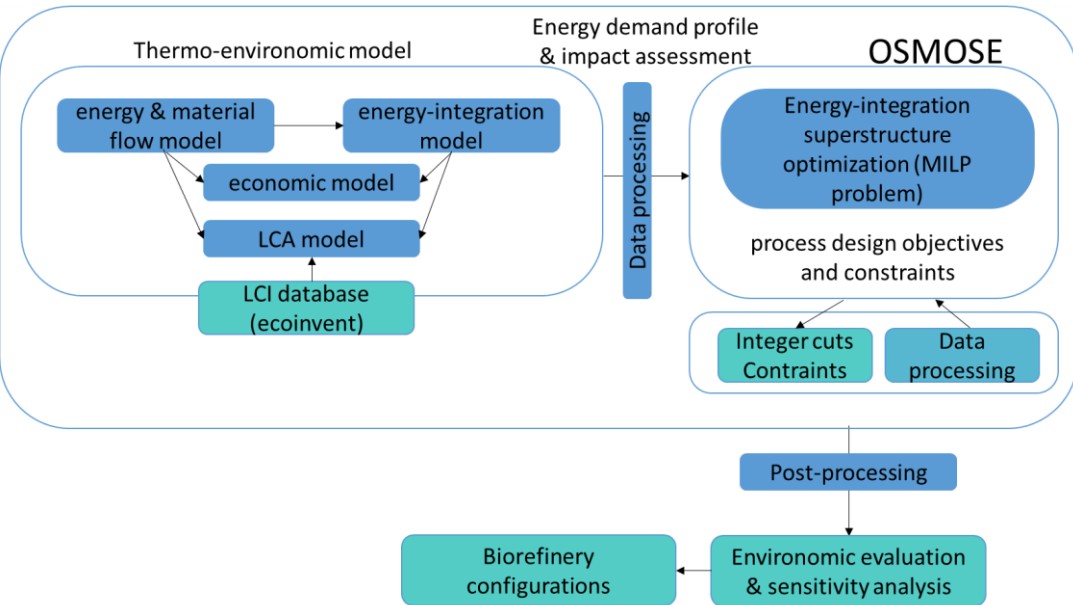

**Figure 6.** Thermo-environomic optimization methodology overview adapted with permission from Ref. [113]. 2017, Elsevier.

Another method linking LCA with data from simulation software was developed by [122] as their simulation tool for water treatment EVALEAU [123]. It automatically imported process data and calculations from the water chemistry calculation software PHREEQC [124] and a water quality database via a Python-scripted code into the LCA Software Umberto. The environmental impacts could then be evaluated through an optimization code in Python. Missing data, e.g., on sludge treatment, were not considered [122]. The EVALEAU framework was also used for comparing the performances of six meta-heuristic algorithms for multi-objective optimization for cost and environmental impact optimization [125]. Another example of dynamic modeling, here in the area of wastewater treatment, is the coupling of the software BioWin [126] through a Python script with LCA in Umberto [127,128]. This script was also used for literature-based complementary calculations (e.g., on energy requirements) as well as assignments between the output flows of the simulation and the flows of the LCA software. A summary of additional formula and Ecoinvent database data used can be found in the Supplementary Material [127], although the Python code could not be found.

Outside the chemical process simulation field, Cerdas (2018) combined engineering models and LCA, where a simulation for the life cycle of electric vehicles (e.g., containing energy mass models for battery systems from the cell up to the battery pack) or mass models for the configuration of the vehicle, as well as driving contain simulation routines for driving cycles is used for so-called integrated computational life cycle engineering (IC-LCE). The concept was implemented in Python and uses Brightway for the LCA calculations. The aim is to use less oversimplification (e.g., without using aggregated environmental impacts from datasets or using different geographical boundaries). Automation is particularly apparent here in the interaction between the different layers (simulation-LCA visualization of results) without the need for user intervention; for instance, the appropriate regional electricity mix is chosen by the software [129].

### 3.6. Process Optimization

Process optimization enables the simultaneous use of LCA and process simulation. Environmental impacts as well as economic considerations are often included in the process evaluation in a standardized way, for example, by means of mathematical programming (e.g., multi-objective optimization), with the purpose of optimization of a fixed process

flowsheet [93,130]. Azapagic and Clift [131] pioneered environmentally friendly process optimization with the life cycle (LCO) approach.

The SUSCAPE framework, which uses life cycle assessment, surrogate modeling, objective reduction, multi-objective optimization and data envelopment analysis (DEA), is a good example of the second layer. A case study on the production of methanol from $CO_2$ and hydrogen assessed eleven CML 2001 LCIA indicators, taking the background data from Ecoinvent database Version 3 [132]. These were included without aggregating the indicators. Building on an Aspen-HYSYS model, decision variables, such as flow rate and temperature, were varied through 1000 samples generated through Latin hypercube design. Thus, 209 of them achieved the desired production rate and were used to build a surrogate model, using 70% for training, 15% for validation and 15% for testing, which was, afterwards, twice re-calibrated adding another 85 and 79 samples generated through a single-objective optimization and a multi-objective genetic algorithm (MOGA). Afterwards, an objective reduction using a single-objective genetic algorithm on the 11 objectives was performed to build the then used initial Pareto frontier. Following this, an MOGA algorithm already implemented in the MATLAB optimization toolbox was used for the multi-objective optimization. The post analysis of the Pareto frontier was performed using DEA, identifying only 10 out of the 126 optimal points as efficient [133]. Although this framework integrates LCA and process simulation simultaneously, and compared to the examples above, evaluates the process in standardized ways, an automated connection between them is lacking. Although the evolutionary algorithm, MOGA, performed well and had attractive properties when integrated with the process simulator, there are numerous problems in constraint integration, flowsheet initialization, decision variables and their boundary selections [133,134]. Pasha et al. [134] recommended using derivative-free optimization algorithms "due to their performance efficiency in discontinuous, non-differentiable, or highly non-linear expressions", such as the evaluation algorithm with taboo list (MODE-TL) used in [135].

Based on the data from the Suscape framework [133], Rodríguez-Vallejo et al. [136] developed a novel DEA approach, tailored to process synthesis and design using a key performance indicator (KPI), thus "preventing unrealistic targets by accounting for thermodynamic limitations represented as mass and energy flow constraints". In their example, they used the GWP and specific cost as KPI and performed the case study in Apentech as well as GAMS for the DEA modeling.

Another tool performing multi-objective process evaluation is LCSoft, which was exclusively developed for evaluation of chemical, petrochemical and biochemical processes and able to integrate "with other process design tools such as sustainable design (Sustain-Pro), economic analysis (ECON)", process simulation and a property database tool (CAPEC DB), allowing for automated data transfer between the packages [128,137]. The tool is based on Visual Basic for applications with an Excel interface and consists of various tools, including LCI knowledge management, a tool for estimation of environmental impact characterization factors using a group contribution method, a tool to perform the LCA calculation and the interface between the different software [137]. Software updates, such as further managing options for LCI databases, new pathways for LCIA calculation and new features, such as parameter sensitivity analysis, normalization, data quality indicator and uncertainty analysis, using Monte Carlo analysis were described and tested by Chavewanmas, Malakul and Gani (2017) and Petchkaewkul, Malakul and Gani (2016) [138,139]. It is not open access but has to be purchased over PSE for SPEED Company Limited [140].

Multi-objective optimization was also performed by Helmdach et al. [128]. This builds an automated connection between the process simulation software gPROMS, the LCA-software Umberto and MATLAB for the optimization. While gPROMS was linked with MATLAB using gO:MATLAB [141], Umberto and MATLAB were connected via Excel and a Java robot mouse emulator, which simulated manual user clicks. Using the TS-EMO algorithm LCA results, as well as costs computed from the mass and energy flows, these can be used to calculate Pareto solutions. A gate-to-gate case study of the

conversion of terpenes derived from biowaste feedstocks into reactive intermediates was performed. Through the passing of all data (e.g., process parameters, material and energy flows) to MATLAB, it is potentially possible to integrate different simulation software and different complex systems, which is, however, limited by much software not exporting gradients, although blackbox optimization also speeds up the calculation time of the optimization [128]. Another example where Umberto was coupled directly via Microsoft Excel with simulation software, namely ChemCAD, can be seen in Denz et al. [142].

### 3.7. Process Synthesis

As an extension of the optimization, a new process design can be created, for example, by selecting options from superstructures or by using evolutionary algorithms. This could influence the unit processes in a process flowsheet, the equipment used or the reaction route for a given product [143–145]. Equation-oriented mathematical optimization problems are not yet integrated in publications for automating the software connections described, but preparatory work is as follows: Using superstructures and solving of the optimization with multi-objective mixed-integer nonlinear programming (MINLP), it was applied, for example, at a hydro-carbon biorefinery [146], for a microalgae-to-biodiesel process [147] and for ammonia production [148]. König et al. [149] used reaction network flux analysis and process network flux analysis to investigate the production of bio- and e-fuels. All of the previously mentioned authors only included the GWP as an environmental indicator. Caldeira et al. [150] used chance-constrained programming (CCP) optimization to identify blend compositions of biodiesel based on a previously published LCA model [151].

A combination of flowsheeting energy integration techniques, LCA and economic performance was performed for ammonia production by Tock et al. [152] based on previous methodologies [114,153,154], stating that the integration of environmental impact likely has an important influence on the engineering decisions, and the choice of its objective function influences the optimal process configuration [153]. The thermo-environomic models were developed in the flowsheeting software Belsim Vali4 [152], and the LCI was integrated as a function of the parameters [153] so that the environmental impact takes the process design and operation into account [152]. The optimization problem of finding the best process layout was solved using an evolutionary algorithm in Matlab [152].

Another approach is, for example, the open-source project BioSTEAM [78], which allows for flexible scenarios and incorporation of uncertainties in the direct interaction between biorefinery design, simulation and technical-economic but also ecological assessment on the basis of the LCA software Brightway2 [79,155]. The Python-based software is for early-stage technologies, especially in the biorefinery sector, and automates agile inventory data and simulations of different scenarios and designs, as well as enabling rigorous uncertainty and sensitivity analyses [155]. Case studies of the software are on sugarcane ethanol production [79], co-production of biodiesel and ethanol from lipid-cane and the production of second-generation ethanol from corn stover [155]. Examples of scientific papers including LCAs using BioSTEAM concern the production from lactic acid production from lignocellulosic biomass [156] and renewable linear alpha-olefins by base-catalyzed dehydration of biologically derived fatty alcohols [157]. Information on BioSTEAM can be found on Github [158] and at the official homepage [78].

Another example of simulation software outside the process simulation, which allow for the automated chance of own values through integrated feedback loops, that has already been tested with LCA is Vensim [159], used by Stasinopoulos et al. [160] for integrating greenhouse gas emissions into the transition to an autonomous vehicle fleet or discrete-event simulation (DES), used by Löfgren and Tillman [161] for manufacturing.

### 3.8. Process Calculations

If available data are not sufficient or the LCA practitioner feels more comfortable with chemical engineering calculations than with process simulation, process calculations can be used. In their classification, Parvatker and Eckelman [19] defined advanced and basic

process simulation as individual methods. However, the distinction between advanced and basic calculations may vary depending on the author. While Smith et al. [32] presented their calculations as simplifications, Parvatker and Eckelman [19] classified them as advanced.

The following is a definition by Parvatker and Eckelman [19], "advanced process calculations build on basic process calculations but include consideration of production scales, equipment efficiencies, heat transfer processes, reflux, and reactor geometries". Comparing this to the case studies on styrene and ABS by Parvatker and Eckelman (2019), the difference to process simulation based LCI is minimal (11 out of 18 LCIA indicators were within a range of 25% compared to actual plant data), and advanced process calculation even performed better than simulation in some categories. In the prediction of GHG emissions and CED, process calculation performed marginally poorer. For the calculation of heat integration opportunities, pinch analysis with tools, such as i-Heat and Aspen Energy Analyzer, was recommended [19]. Frameworks for calculations are available, such as the scale-up framework for chemical processes by Piccinno et al. [162].

In contrast to advanced process calculations, fixed values for temperature, pressure, etc., are applied for the basic calculation technique, which do not vary over time. Mass and energy equations are typically used. However, since time must also be invested in research for assumptions about, e.g., process yields and solvent recovery, Parvatker and Eckelman [19] recommended performing an advanced calculation immediately if possible.

As an example of basic calculations, Jiménez-González, Kim and Overcash [163] developed a methodology for developing gate-to-gate LCI information, including rules of thumb, stating that these rules are dynamic and subject to change owing to current trends in industrial production processes, improvements and pollution prevention methods.

Falling into the process calculation category, an approach using advanced process calculation combined with a molecular descriptor was taken by Kleinekorte et al. (2019), who developed a fully automated neural-network-based framework, aiming to integrate process descriptors to the molecular properties to be able to distinguish different production pathways when producing a chemical. The framework includes feature selection through stepwise regression, setup of the network architecture using a genetic algorithm (GA) from MATLAB and predicts 17 environmental impact categories: the Recipev1.08 (H) midpoint categories [143]. For a case study comparing component- (solely based on molecular descriptors) vs. process-specific networks, the training data used were 63 organic chemicals obtained from the Gabi database, and 10% of them were chosen randomly for the validation set. To avoid overfitting and due to limited input data, the feature selection only chooses 10% of the number of training data as the final number for the descriptors from originally 178 molecular and 7 process descriptors. The process descriptors were based on Patel et al. [164] and include, for example, "the concentration of each component at the reactor outlet [ … ] assuming ideal phase and chemical equilibrium", which "is assumed to correlate to the required separation effort" [143]. Compared to the molecular-based ANN, the process-specific network increased the prediction in 10 of 17 midpoint indicators, although the absolute coefficient of determinations was small, with values lower than 0.4. for 16 impact categories. Compared to the literature, the ANN created by Wernet et al. [165,166] performed worse in 14 of 17 cases. The authors also stated that they performed similar to Song, Keller and Suh [167] with their $R_2$ for GWP. However, when comparing their acidification with the other neural networks (NNs) using the conversion factors for impact assessment methods from Dong et al. [168], they performed significantly worse. Kleinekorte et al. [143] additionally performed a case study on the $CO_2$-based pathway for formic acid and methanol and found when comparing it to LCAs from the literature that the potentials are overestimated by their process-specific ANN, but the trends are predicted correctly.

A comparison of the coefficients of determination of the NNs can be seen in Table 1. It should be noted that this model is a mixture of molecular-structure-based model and process calculation and is, therefore, not a straightforward automation of the latter.

**Table 1.** Comparison of molecular-descriptor-based prediction on LCIA using neural networks and Gaussian process regression; MLR = multiple linear regression, ANN = Artificial Neural Network, GPR = Gaussian Process Regression, CED = cumulative energy demand, GWP = Global Warming Potential, EI99 = EcoIndicator 99 Total, (T) A = Terrestrial Acidification; for molecular descriptor based prediction on LCIA using Neural Networks.

| | | Finechem | Predictive_LCIA | Process-Specific Environmental Impacts | |
|---|---|---|---|---|---|
| | Using | R (available as R-Package) | Tensorflow and Brightway2 (available on Github) [169] | Genetic algorithm from the Matlab toolbox | ANN vs. Gaussian Process Regression |
| | Developed by | [166] | [165] | [167] | [143] | [170] |
| | Model | ANN, (MLR) | ANN | ANN | ANN | GPR, ANN |
| | Input data | 103 chemicals (58 from Ecoinvent v1.3 & 45 inhouse data) 'basic chemicals to solvents, chelating agents and pesticides | 392 chemicals (96 Ecoinvent & 296 inhouse & industry data) | 166 organic chemicals from Ecoinvent | 63 organic chemicals from the GaBi Database | Three datasets ranging from 62 to 547 organic chemicals |
| | Training/ Testing/ Validation Splitting in % | 85/15/0 | 85/15/0 | 84/6/10 | 85/5/10 | 80/10/10 using Kullback-Leibler divergence |
| | Molecular Descriptors | 2,6,17 | 10, reduced by PCA to 8 factors | 3839 decriptors reduced by PCA to 60 features | Based on Wernet [165,166] architecture 178 | 178 & 7 process descriptors |
| Prediction Accuracy (Mean relative error [%]) | CED | 22.4–28.8 | 29.1 | 30 | | |
| | GWP | | 58.2 | 65 | | |
| | EI99 | | 20.7 | 21 | | |
| | (T)A | | | 26 | | |
| Coefficients of Determination for the Impact Indicators | CED | | 0.58 | 0.45 | | |
| | GWP | | 0.41 | 0.48 | ~0.08 | ~0.23 | ~0.31 |
| | EI99 | | 0.69 | 0.87 | | |
| | (T)A | | | 0.73 | ~0.02 | ~0.00 | ~0.17 |
| | Application | petrochemical field [171,172]. postcombustion $CO_2$ capture [173] bio-based production of platform chemicals [174,175]. Soft PVC plastic and a tobacco flavor [176] | | Organic chemicals Case studies acetic anhydride hexafluoroethane [167] | $CO_2$-based production of methanol and formic acid to their respective fossil pro- duction pathway | / |

### 3.9. Stoichiometry

Stoichiometric methods, which are based on balanced chemical reaction and molecular mass, are easily available. They are the basis of the calculations and simulations mentioned above and have performed individually reduced time requirements but lack information and process configuration, production scale, plant design or separation processes, which leads to the ignorance of the important factors leading to environmental impacts. They can be supplemented by thermodynamic data, such as heat of reaction and heat capacities, to estimate the heat requirements and estimations of the separation of the target product from the by-products after the reaction, leading to three levels. Level one only includes the balance reaction, level two adds the heat of the reaction and level three incorporates the yield [19].

Parvatker and Eckelman's [19] calculations for the different process levels showed that the three stoichiometry levels were more likely to underestimate the environmental impact compared to the real plant data (while process calculation and process simulation were more likely to overestimate it), and that in 10 of 18 recipe midpoint categories, the results became closer the higher the stoichiometry level. However, in five indicators, a reversed effect could

be seen, where the accuracy from level one to three decreased. The method could be useful for a comparison of products or processes for LCA practitioners with time constraints.

When screening the literature, a comparison of biofuel- and carbon-based fuel pathways, including the global warming potentials, was found by König et al. [149]. Their calculation, however, is based on two screening methodologies for early-stage development, the process network flux analysis (PNFA) [177] and the reaction network flux analysis (RNFA) [178]. Although the RNFA is based on stoichiometry, the yield of the reaction and the component properties, such as molar mass and enthalpy, the GWP calculation still needs more detailed data for the PNFA, such as the effect of solvents, concentrations and separations [149,177].

For the detailed literature analysis in this paper, only one publication with automation potential could be identified. Pereira et al. [37] developed two models for estimating the steam consumption in the production of chemicals based on different predictor variables such as reaction type. The models, using probability density functions (PDFs) and classification trees, were derived from "production data provided by a consortium of industrial partners representing leading companies in fine chemical and pharmaceutical" and a case study in which the intermediate substance 4-(2-methoxyethyl)-phenol, derived from different synthesis routes, was tested. Steam consumption was chosen as it is usually the highest energy utility consumption in batch plants [37]. As predictor variables, reaction type and reaction mechanism, presence or absence of distillation process and reflux condition, maximal temperature and time required for heating as well as process mass intensity and steam consumption during distillation process were used, it can be argued that this is a combination of process calculations and stoichiometry.

Including uncertainty calculations using Monte Carlos, the model outputs are in the form of intervals rather than point estimations, and it is stated that estimation errors can also occur due to the simplifying hypothesis that energy utility consumption is mostly dependent on the production processes rather than on specific reactants, auxiliaries and products [37]. No further use nor source code for the model could be found, which limits the gain in knowledge and further development by other scientists.

### 3.10. Molecular-Structure-Based Models

The physical–chemical properties of chemicals, including molecular weight, composition and functional group, contain information about energy and resource requirements for production, which are extensively used in combination with advanced statistical techniques for prediction of toxicity [19,166], as well as in other areas, such as bioinformatics, medicinal chemistry or chemometrics [179]. A use for LCA was explored by various authors, as described below. The prediction quality is, however, in the context of a narrow set of chemicals used for training the model. Additionally, they are used to directly calculate a predefined selection of LCIA results, not to generate LCI data. The models do not show mechanical relationships other than statistical ones, which makes testing different process configurations impossible [19]. However, they are a good alternative for screening purposes, and in a case study by Parvatker and Eckelman [19], the performance was similar to that of their stoichiometry calculations at level three.

Wernet et al. [166] assessed the capabilities of multiple linear regression (applying a least-squares analysis) and neural network models, or more precisely, multilayer perceptrons (MLPs) for this purpose. The models created were based on the inventory data of 103 chemicals, ranging from basic chemicals to solvents, chelating agents and pesticides. The data quality is rated as generally high—58 of the datasets were taken from the Ecoinvent database v1.3 (1) and 45 were created in house by detailed analysis of production processes. The MLPs used a Broyden–Fletcher–Goldfarb–Shanno algorithm, a quasi-Newton method, for training 85% of the data on one input, one hidden and one outer layer, with an optimized number of neurons in the hidden layer and normalized data. Further, 30 combinations of training and corresponding test sets containing 15% of the data were created, and the test set excluded a number of quantifiably unique chemicals.

Further analyses of the relative prediction errors (MRE) of the NNs showed that the detailed and the condensed model have comparable prediction capabilities (22.4% and 22.9%) while the minimum model consults in a higher MRE from 28.8%. Additionally, the authors created new NN models through leave-one-out (LOO) cross-validation, which showed "that even a relatively small increase in training set size has a considerable impact". The LOO model scored a mean relative prediction error of about 6% and a q2 of about 0.8, which confirms the validity of the approach and the usefulness of the models.

The NN model performed better than the corresponding linear regression models, showing that a linear relationship is not adequate to describe the model. Even the MLP with the minimal input set almost always had a higher mean Pearson and Spearman rank correlation coefficient (R2) than all the linear regression models.

They calculate the impact factors by focusing on the cumulative energy demand (CED), as well as the global warming potential (GWP), the biological and chemical oxygen demands (BODs, CODs), the total organic carbon (TOC) and the eco-indicators total (EI99 T), Human Health (EI99 HH), Ecosystem Quality (EEI99 EQ) and Resources (EI99 R).

As a limitation, the authors state "that the current models are lacking in range and the results carry a level of uncertainty" [166].

For further development with a focus on reducing overfitting, they concluded that "more advanced techniques involving soft or hard pruning of the network weights could be utilized to reduce the overfitting problem. However, it is expected that the impact of a richer database would be much more significant" [166]. The richer database is also recommended by the authors because a loss of predictive capabilities is seen due to the limited capacity of NNs to extrapolate. To be able to interpolate "the training data should at least broadly cover the chemical range of the test data", as unique and untypical chemicals in the test set revealed the problem of limited generalization capability [166].

Their subsequent article, which already used 392 chemicals to train an NN, was compiled in 2009. Here, 296 were cradle-to-gate chemical data from industry partners and 96 from Ecoinvent. The data quality was rated as high, and everything was based on the average European (UCTE) electricity mix. For usage in the LCI, the model estimates direct heat and electricity use but also calculates LCIA results, such as the CED, the GWP and Ecoindicator 99 [165].

As input, they selected ten descriptors from a former larger group. These were then reduced to eight factors, accounting for over 95% of the variance, by performing a principal component analysis (PCA) to reduce the trade-off between communicating as much information as possible to the model while minimizing input size. Using seven hidden neurons, resulting in 71 weights in the NN and a data-to-weight ratio of 4.8. (indicating some overfitting), the model was trained with a Broyden–Fletcher–Goldfarb–Shanno algorithm, a quasi-Newton method. Uncertainty was incorporated as the final model puts out 30 results from the 30 best-performing models, which can be used for uncertainty analysis. The coefficient of determination ($q^2$) for those models was usually between 0.4 and 0.7, and the electricity and heat prediction model, as well as the CED and the Ecoindicator99 predictions, perform well, while the GWP models perform less well due to a lower coefficient of determination and a higher variation in the original data [165].

Based on these two publications, the R-Packet FineChem was developed to use this NN for the generation of LCI and LCIA data in the petrochemical field, which helps practitioners obtain LCI and LCIA data through the use of a model in an automated way, by only needing to know the molecular structure [171,172]. An example where FineChem was used is the calculation of LCA aspects of solvent selection for postcombustion $CO_2$ capture [173,180], were it was used for creating a computer-aided molecular design (CAMD), which helps designing or finding different chemical compound/s, such as solvents, polymers and pharmaceutics, by using a large number of feasible molecular structures based on the estimation of desired physiochemical properties [181]. Integrating FineChem into a CAMD framework would allow for integrating LCA criteria during the solvent selection in the first stage compared to the post-assessment in FineChem, which has to be conducted in a second

step [172]. Another example of where it was used is the evaluation of bio-based production of platform chemicals [174,175]. Limitations of these LCAs in FineChem as well as an outlook on developing FineChem to a group contribution-based version were described by [172]. Wernet's ANN [165] can also be used for free via the online tool EstiMol [182,183].

Using multiple hidden layers in contrast to FineChem [167] also generated an ANN to predict LCIA results on the behavior of molecular structure. The model development was based on three groups (training, validation and testing) using "166 unit process datasets for organic chemicals from the Ecoinvent" database. The NN was created using the Google Tensorflow framework in Python 2.7, and around 4000 molecular descriptors were calculated for each chemical by Dragon7. To reduce the number of descriptors, filter-based feature selection and PCA (preserving 95% of the variances in the original dataset) were tested as well as using all descriptors as model inputs. After testing them on 72 different model settings (6 impact categories, 3 levels of hidden layers and 4 levels of hidden neurons), PCA was further employed to reduce dimensions, as it had, in general, higher R2 values at the 72 tested model settings. Six models were created for each model output and the LCIA categorized cumulative energy demand (CED), global warming (IPCC 2007), acidification (TRACI), human health (Impact2000+), ecosystem quality (Impact2000+) and eco-indicator 99 (I,I, total). Whereas the models for acidification, EI99 and human health "perform relatively well, with R2 values of 0.73, 0.87 and 0.71," the CED, ecosystem quality and the GWP had a lower performance with 0.45, 0.48 and 0.21–0.31. Performance can be compared to Wernet's [165] model in Table 1 [165,167]. A drawback in this paper is that the model was only compared with the first Wernet model [166] in the paper, instead of comparing it with the more comprehensive model from 2009 [165], where the $R^2$ data were also available. There are also two case studies in the paper using the chemicals acetic anhydride and hexafluoroethane (HFE) [167]. In general, chemicals with a high life cycle impact can be identified as having higher estimation errors due to the limited training data available for them. Additionally, those chemicals with especially high CED are mostly pharmaceuticals, and the energy demand through selectivity and purity requirement in this industry would need more specific training data.

Assumptions and uncertainties in the initial training data from the Ecoinvent database should be considered when using this model. Additionally, this study used the Euclidean distance-based AD measurement to characterize the uncertainty estimation, which provides an indication of prediction errors. In their outlook, the authors stated that a combination of the AD with methods such as nonparametric probability density distribution could improve the uncertainty information and that future research should consider synthesis pathway descriptors, such as reaction temperature, existence of catalyst or reaction selectivity, next to the molecular descriptors [167]. The model can be accessed on Github [169].

Following on the ANN model mentioned in the Process Calculation section above [143], Kleinekorte et al. [170] presented a comparison between the ANN model and Gaussian process regression (GPR) for limited training data at a conference. They concluded that "the GPR outperforms the ANN in terms of prediction accuracy on all datasets". As GPR is considered to be a powerful machine learning regression algorithm, especially also useable for small datasets [184], a literature search was obtained on its usage for LCI data generation so far. No further usage in combination with molecular models was found, although in research on usage for predicting missing temporal or geographical data [185] and for energy prediction during wheat production, the authors concluded the ANN model with radial bias function was more accurate [186].

Meyer et al. (2019a) combined data-mined data from 348 facilities on air emission and production volume with physicochemical property data, such as molecular weight, vapor pressure, density and water solubility. Extracting the data for 45 chemicals first using standardized emission and waste inventories (StEWI) and a Python script, as well as ChemSTEER, they then created, in R-Studio, a single classification and regression tree (CART) and, in an extension of the CART approach, a random forest (RF) model, with 2000 trees, which should increase the prediction accuracy and purpose-driven reconcili-

ation of approaches to estimate chemical releases. However, when comparing the error parameters, the CART model has better results for predicting new chemicals or chemicals with poor data (chemicals outside the test and training set). Both models would need a larger curated training set, but if a good model is generated, those using the model need little knowledge about the process, and data quality can be good when current and high-quality data are used for the training. Both R-codes are available in the supporting information for the publication [33].

Hou et al. [187] estimated missing ecotoxicity characterization factors for 2122 USEtox chemicals using fourteen physical–chemical characteristics obtained by the open-source software OPERA (Open Quantitative Structure–activity/property Relationship App) [188,189] and by using the Python Scikit-learn machine learning module [190], which is an open-source Python library, including various regression, classification and clustering algorithms. In their example, they calculated the hazardous concentration 50% (HC50) as a regression problem through following machine learning methods: "k nearest neighbors (KNN), support vector machine (SVM), neural networks (NN), random forests (RF), adaptive boosting (AdaBoost), and gradient boosting machine (GBM)". They also compared the performance with traditional QSAR models, such as ECOSAR, "ordinary least squares regression (OLS), partial least squares regression (PLS), and principal component regression" [187]. The machine learning models had better prediction performance than the traditional QSAR models and, in total, RF had the best predictive performance. Based on these results, the HC50 and CFeco were calculated for 552 chemicals without existing values using an RF with 1000 trees and 4 as maximum feature for splitting trees. Uncertainty was addressed by removing outliers with Cook's distance measure and by giving a confidence interval for the estimated data. Ref. [187] emphasized that their results are only useful references as long as chemical-specific laboratory tests are not available. The limitations of their approach are the difficulties in interpretation due to the missing explicit mathematical function to explain the RF model ("black box"), the fact that the training causes high computational costs and variables with less unique values are handled as less important. An extension with more physical–chemical properties is also being considered. The advantages of their approach are the time savings and the good performance, as well as the ability to easily tune parameters and the wide applicability [187]. The Python code directly linking to this article could not be found on the authors' Github page [191].

Mio et al. (2021) created an extension to multiscale molecular modeling in their LCA to identify a suitable material for a set of cylinder head covers for a marine engine, where they compared aluminum with different nano-engineered thermoplastic polymers. As visualized in Figure 7, the molecular modeling was applied on different modeling scales.

Although this extended approach is described here, as it is interesting to incorporate LCI in all phases, the paper lacks methodological detail.

### 3.11. Proxy

The use of proxies, e.g., from existing databases, is a popular approach that reduces data search time [19,176,193] but requires expert knowledge to establish the similarities in any process routes and synthesis conditions [19,193]. Canals et al. [193] differentiated four proxy data types for biobased products: scaled proxies (through linear scaling), direct proxies (data with similar characteristics and functions), average proxies (average of several similar products) and extrapolated data (through extrapolating similar LCIs). After comparing several case studies, they concluded that quality as well as effort increases when moving from scaled to direct to averaged and then to extrapolated, although sensitivity analyses should be carried out for all of them. Parvatker and Eckelman (2019) maintained, after performing their case studies on Styrene and its downstream product ABS, that its proxy data should be mainly used as a placeholder, to be later replaced with data from higher areas of the pyramid (see Figure 2). However, both studies concluded that using a proxy is better than omitting the information [19,193]. A proxy method used by Parvatker and Eckelman [19] with automation potential is monetary-based, using chemicals with similar market price, although the LCIA

results were not as good as those from the direct proxy. Examples of using an average and extrapolated proxy can be seen in Wernet et al. [176], who also concluded in their two case studies that, compared to the process model approach or the use of the Finechem tool [171], the use of proxies is not recommended.

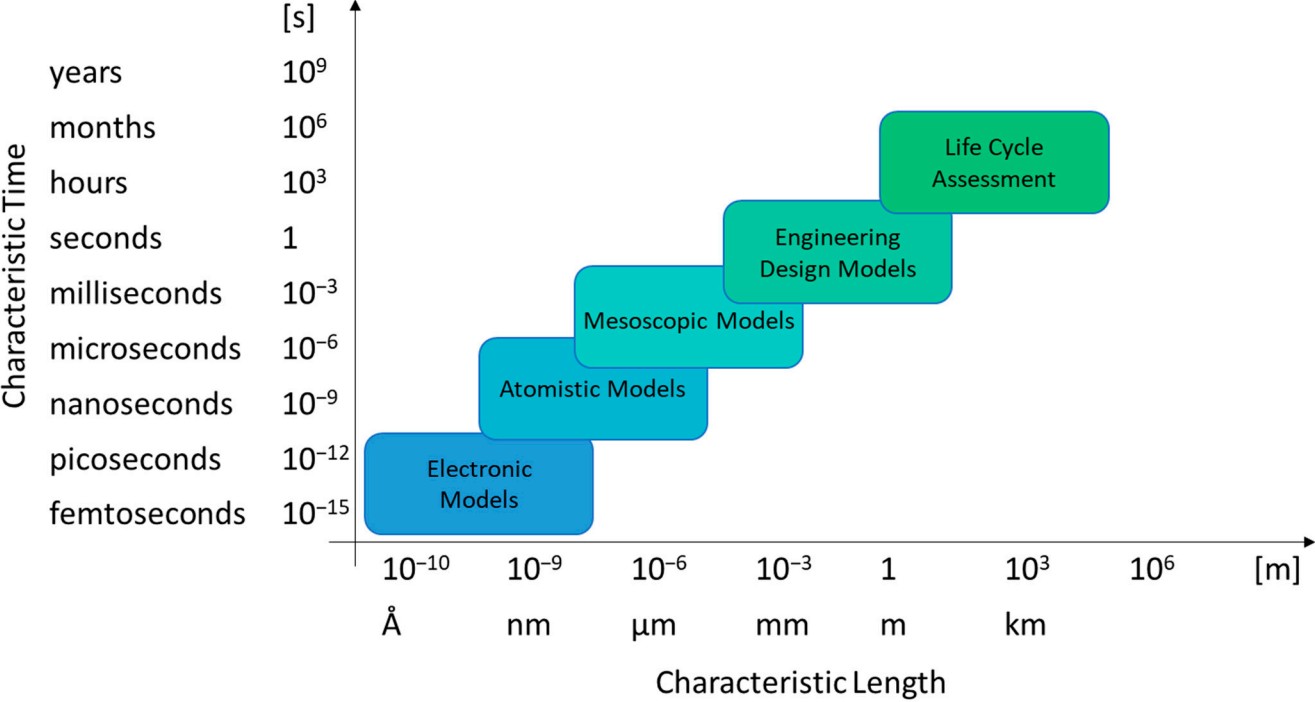

**Figure 7.** Multiscale molecular modeling scheme adapted with permission from Ref. [186]. 2021, Elsevier. "The axes report an approximate range of characteristic time and length scales covered by specific modeling techniques. Overlapping areas allow information exchange between models, which is usually needed for parameterization of higher scale simulations or for gaining a detailed resolution of selected portions of bigger systems" [192].

Developments in using proxy data for the direct prediction of LCIA results can be seen in Steinmann et al. [194], who developed a regression-based framework for coal power plants, and in Pascual-González et al. [195], who used multi-linear regression and mixed-integer linear programming.

An approach used for our detailed literature analysis of automation potential for LCI was performed by Hou et al. [196]. Their automated approach fills in data gaps in unit process (e.g., energy demand for production) through a similarity-based link prediction method, which identifies similar processes via data comparison of other intermediate and elementary flows and estimates missing values by averaging the corresponding data of the similar processes. For their example, they used 2546 processes from the Ecoinvent 3.1. database and built a matrix using 7029 flow data from these processes. Comparing approaches with and without normalization, they were able to predict missing values with high accuracy when using no normalization with 1% and 5% missing flows. They provided two case studies on diesel, one for a dataset of another database (U.S. LCI database) and one for a dataset directly from Ecoinvent [196]. Building on the work of Xu et al. [197], the authors extended their research with a decision-tree-based supervised learning approach to improve their forecast on identifying zero and non-zero flows. Their code can be accessed via GitHub [198].

### 3.12. Bibliometric Analysis

Applying mathematical and statistical methods to books or other media of communication was first defined as bibliometric by Pritchard [199]. In our research, we apply it to

the above-identified and discussed literature on automation in LCA to identify relationships, trends and patterns (e.g., countries with higher research in this area, journals that publish on these topics and keywords). We, therefore, identified the key publications for automation, listed in the Supplementary Material in Table S3. When there was more than one interlinked paper (e.g., on a method or software development), only one paper from these researchers was selected. For example, Wernet published a pre-study for his final ANN used in FineChem in 2008 [166]. This was not included and only the 2009 paper was used [165].

Research Rabbit [24], VOSViewer [22], Web of Science (Analyzing Tools and Citation Report) [23] and Microsoft Excel were used for the different analyses.

Out of 30 closely analyzed literature sources, 10 were published by US first authors and 20 by European authors. Including all authors (using Web of Science the analysis), 17 countries with 45 regions were counted in total, where 11 regions were counted from authors from US institutes and 28 from 10 European countries (Supplementary Material: Table S1). Authors with the highest number of papers (four papers each, including co-authorship) are Belaud JP, Hungerbuhler K, Ingwersen WW, Meyer DE and Papadokonstantakis S (Supplementary Material: Table S2).

As demonstrated in Figure 8 in Europe, the topics are widely spread over the LCI data-generating areas. However, 11 papers deal with process simulation. The US governmental focus is very much on database improvement with three publications published by a first author from the US EPA. Individual research institute focus could be not clearly identified; only three publications on process simulations were from the Université de Toulouse in France and two by the Ecole Polytechnique Federale de Lausanne in Switzerland. Most studies were analyzed in the sector of process simulation with 12 literature sources, followed by molecular-structure-based models with seven and LCI Databases with six papers.

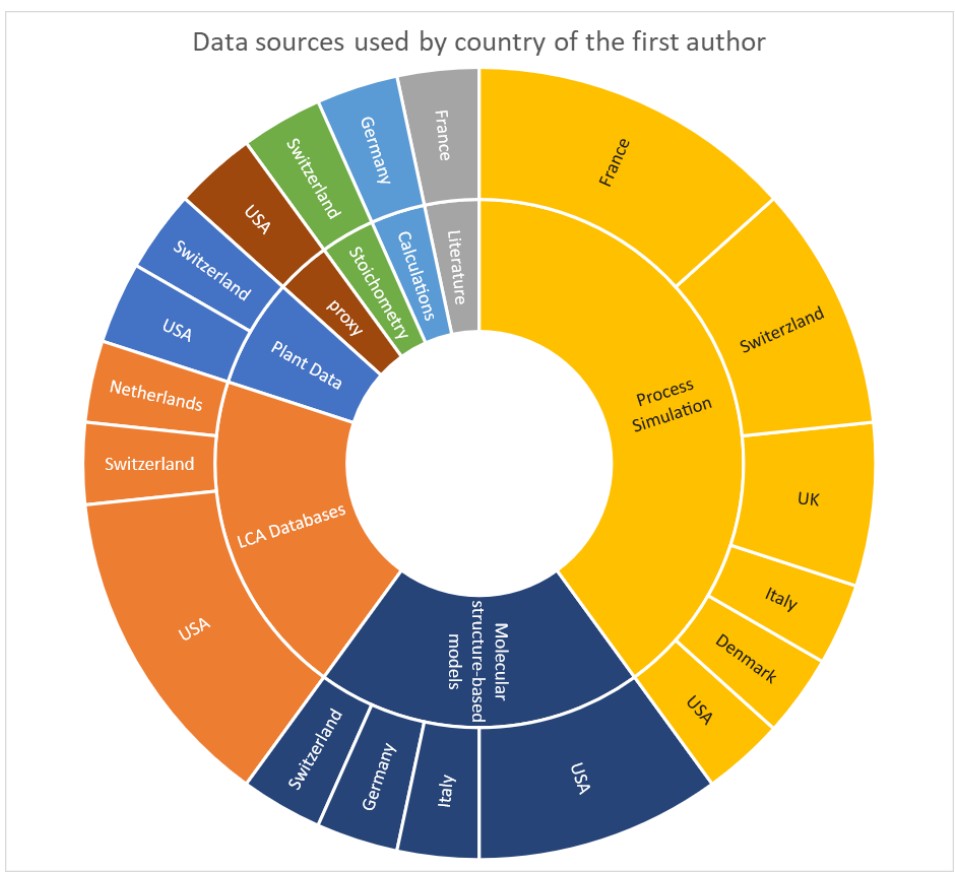

**Figure 8.** Evaluated literature by category and country. The inner circle shows the LCI data-obtaining categories, while the outer circle shows the country of the employing institute of the first author. Process Calculation is shortened to Calculation and Scientific Literature to Literature.

Analyzing the journals used for the publications on these topics, six papers were published in the *International Journal of Life Cycle Assessment* (Springer; 5.553 five-year impact factor), four in *Sustainable Chemistry & Engineering* (American Chemical Society; 8.471 five-year impact factor) and two in *Chemical Engineering Transaction* (Italian Association of Chemical Engineering) [200]. The other 18 papers were each published in different journals.

Most papers, eight in total, were published by the American Chemical Society, seven in Springer journals, six in Elsevier journals and two each by the publisher Royal Society of Chemistry and Italian Association of Chemical Engineering, respectively.

The number of publications rose slightly when looking at the trend between the first published paper in 2009 and 2021, as well as the citation. However, as the increase in citations can also be retraced to the higher number of papers, the grey line in Figure 9 normalizes it by dividing the citations by the published papers until that year. This shows that there is no significant increase in citations in this area.

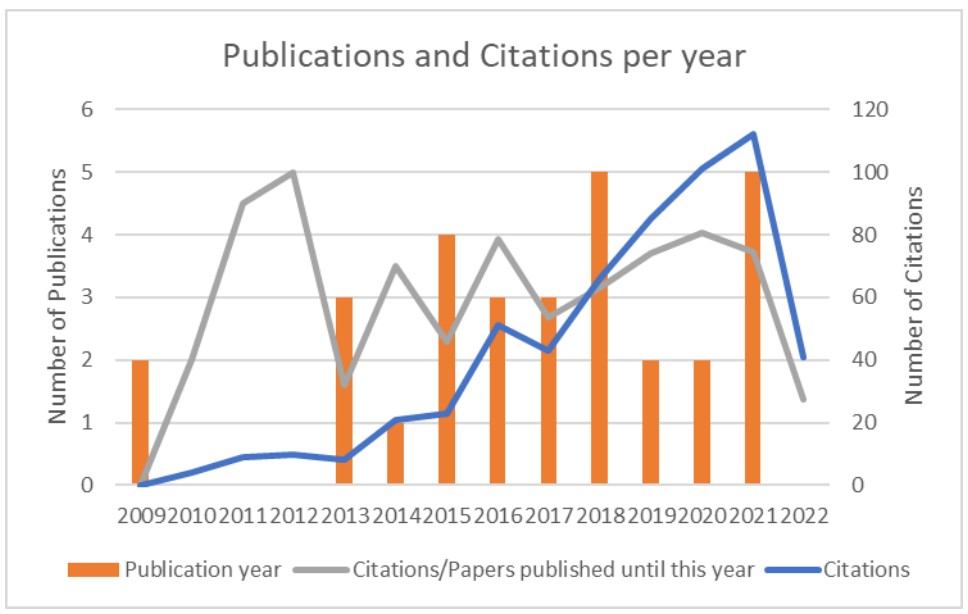

**Figure 9.** Citations on the reviewed publications (analyzed in Web of Science). Orange = Analyzed literature per publishing year; grey = Citations per year divided by until this year (including this year) number of published papers; blue = citation of the analyzed literature per year.

Analyzing the papers which took up aspects of automation on the research paper visualization platform Research Rabbit [24] showed connections between many universities, especially when looking at the author relations, mainly via collaborations between university professors (e.g., Olivier Jolliet, André Bardow or Maurizio Fermeglia; see Supplementary Material: Figure S1). Some institutes do not show connections in this context outside their country, e.g., researchers from the University of Toulouse or the University of Illinois at Urbana-Champaign [201].

Sorting papers by category, Web of Science rated 16 papers as environmental science, 14 papers as environmental engineering and 12 as chemical engineering (detailed Figure S2 with more categories in the Supplementary Material). As there were only 30 papers, it seems that double or multiple counting was used.

The investigation in VOS Viewer, which examines word frequencies in the titles and abstracts of papers, did not reveal any usable trends after words, such as 'process' (average popular 2016), 'data' (2016,5), 'model' (2017) or 'approach' (2018). Despite variation in the settings, the results reflect the general topic. It would probably be necessary to search a larger number of papers. The figure can be found in the Supplementary Material, Figure S3.

### 3.13. Empirical Analysis

As the categories for the following analyses were not defined automatically by the papers metadata but identified by the content of the studies, this section is defined as empirical analysis. It consists of an analysis of research fields or products the LCA focuses on, an analysis of usage regarding TRL and development status of the method/software in the papers as well as an analysis regarding methods used for automation and open science aspects fulfilled.

An individual analysis of research fields or products addressed in the papers (Figure 10) shows that the highest number (14 of 30) focuses on chemical products, especially in the areas of molecular-structure-based models and process simulation, whereas especially database topics and the usage of proxy data are more generally applicable. Based on examples and recommendations in the various papers, the usage of the different data sources regarding application to different TRL levels (x-axis) is visualized in Figure 11. Additionally, on the y-axis, the papers were also sorted by the degree of development of their method or model, which was defined by data obtained from the content as well as from research on further usage (details can be seen in the Supplementary Material: Table S3). Only for the categories process simulation, LCA Database and Molecular Modeling are methods implemented in software environments for easy application and only in the overlapping categories "Database" and "Process Simulations" were methods and models used after the initial papers. For all other LCI sources (plant data, scientific literature, process calculation, stoichiometry and proxy data), no usage in other papers could be found and no software implementation.

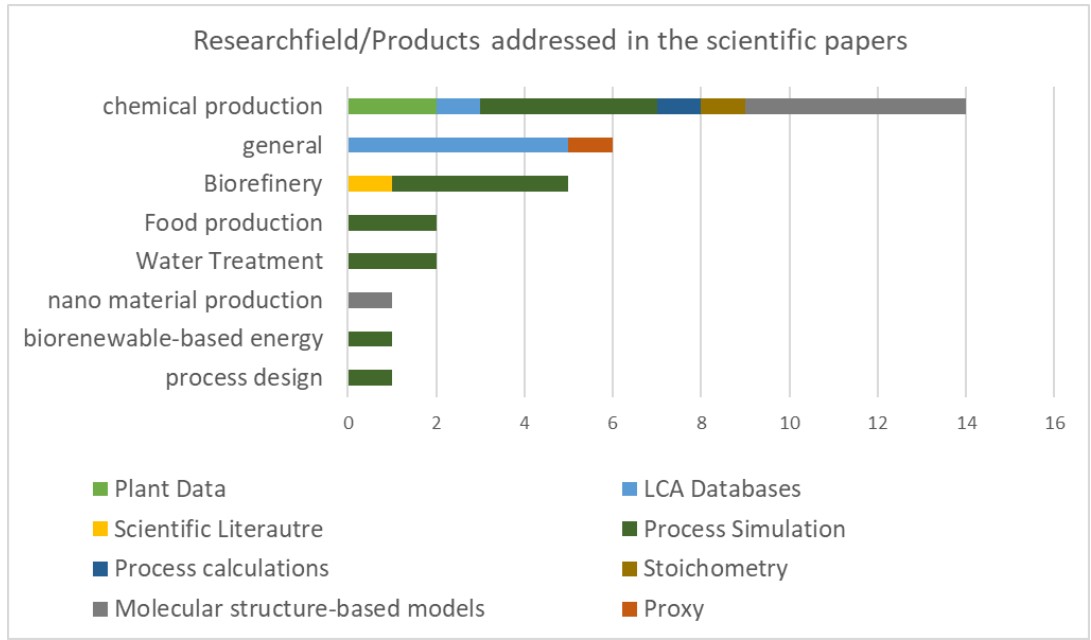

**Figure 10.** Products addressed in the LCA case studies or aim of the study by LCI-generating category (as some papers address more topics, the number is higher than the total number of papers (*n* = 30)).

Stoichiometry and molecular structure models can be used until TRL 9, though this is not recommended, as process configurations and plant design are not considered. Proxy data should only be used as placeholder and replaced later [19].

### 3.14. Open Access and Open Source

Analyzed with Web of Science, eleven papers of thirty were open access, from which five had the status green accepted, four were gold hybrid and three were green published and green submitted. Only one had gold status. The other 19 records did not contain data in the field being analyzed.

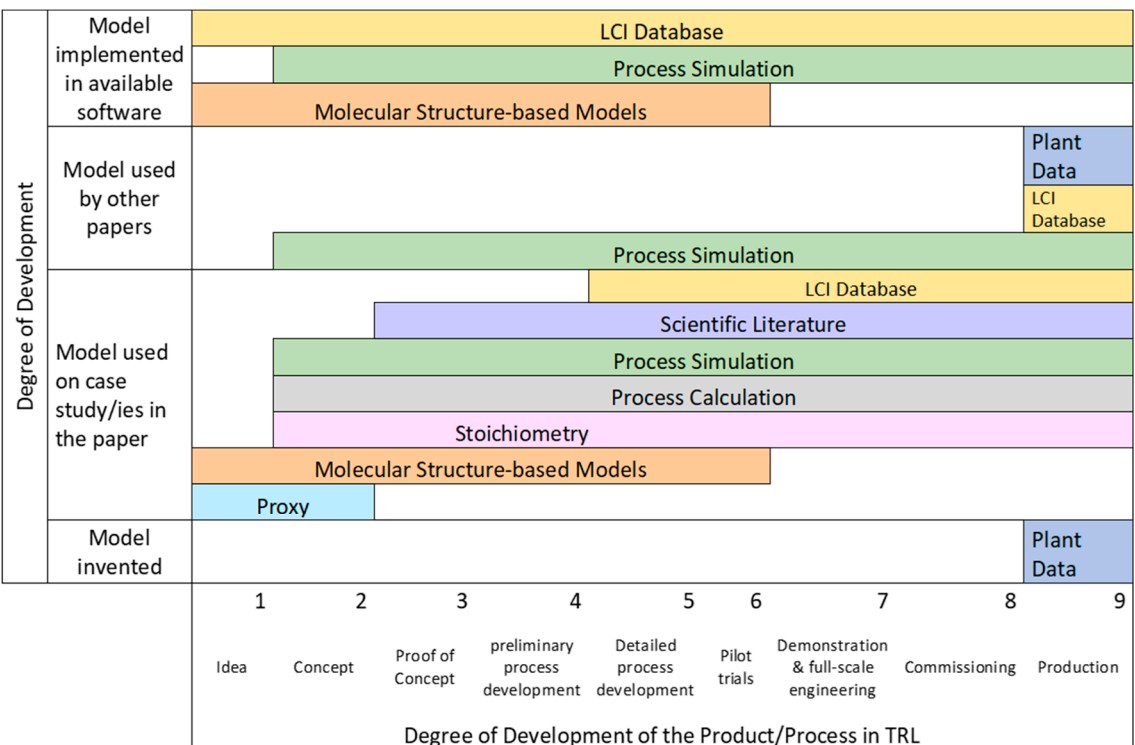

**Figure 11.** LCI data generation method used in the reviewed papers per Technology Readiness Level for the Chemical Industry (TRL, defined in [202]) and degree of development of the method/software. Color coding is per LCI data generation method. Stoichiometry and molecular structure models can be used until TRL 9, though this is not recommended, as process configurations and plant design are not considered. References for the different columns are the following. Model implemented in available software: LCI database [55,67,68,76,77], Process Simulation [79,155], Molecular Structure-based Models [165,166]; Model used by other papers: Plant Data [31], LCI Database [41], Process Simulation [107,123,135,139]; Model used on case study/ies in the paper: LCI Database [56], Scientific Literature [80,83–85], Process Simulation [100,101,110–114,128,129,154], Process Calculation [145], Stoichiometry [37], Molecular Structure-based Models [33,169,172,189,194], Proxy [198,199]; Model invented: Plant data [36].

Regarding the availability of the software or source code to use the method developed and described in the paper as an unrelated researcher, the source code for open access could only be found in six papers, mainly saved on Github.com or available as software extensions (BioSTEAM [78], R-Package Finechem [173]), while four authors stated in their publication that it could be obtained upon request. For most papers—nineteen in total—the source code or software could not be obtained or found online. Of these, where the code or software is open access, still no further usage from unrelated researchers was found—follow-up papers using the software/method always contained a primary author.

### 3.15. Uncertainty

Parameter uncertainty was incorporated in ten papers, in which a Monte Carlo analysis was performed four times and twice referred to the pedigree matrix. For the molecular-based models and proxy data, different kinds of distance measurements/intervals were used and were integrated once for plant data fuzzy intervals [36]. Other approaches, such as combining fuzzy logic and Bayesian networks, as proposed by, e.g., Riali, Fareh and Bouarfa (2019) [203] for dealing with ontologies, were not found.

The listing per paper can be found in the Supplementary Material in Table S3.

### 3.16. Further Use of the Models

For 19 papers, no usage of the developed model/framework after publication could be identified.

For those with examples outside the scope of the original paper and further usage in other publications, such as the FineChem tool by Wernet et al. [165,166], the examples were created with contributions from prior involved scientists, such as Papadokonstantakis or Hungerbrühler. This is also the case with BioSTEAM, where most papers were published by staff from the University of Illinois at Urbana−Champaign, although in the paper, a collaboration seems to have been established between the developing university and the University of Wisconsin–Madison [157]. Other examples where the developers are contributors to their own models include the Suscape-Framework [133], LCSoft [137], EVALEAU [122], Ecodesign Framework [103] and Knowledge Engineering (KE) for data extraction [81].

Integrating software developers seems to benefit real-world application tools, as in Kuczenski et al. [62], where two developed frameworks for data connection can be used through the OpenLCA Olca and Brightway.

### 3.17. Automation Techniques Used

The most popular methods which favor automation were connections between software, such as those building interfaces or including LCAs in existing software (all process simulation LCAs (10–21 in Table S3 in the Supplementary Material), ANNs (for molecular modeling—22, 24–26 in Table S3), techniques used for machine learning, such as probability density functions, random forest or Gaussian process regression (23, 26–28 in the before mentioned Table S3), as well as techniques to connect or search for LCI data, such as using ontologies for linking data (2,4,5,7–9 in Table S3). Data extraction with knowledge engineering and data mining using linked open data were each also used once (2,9 in Table S3).

### 3.18. Artificial Intelligence Techniques Used

Automated learning of models is a central theme in classical (inferential) statistics but more recent in artificial intelligence (AI) and its subcategory machine learning [204]. How often it was used in the reviewed papers is analyzed here. In contrast to traditional statistical methods, AI can be used to obtain solutions without prior knowledge of relationships and dependencies. The techniques build on emulating biological processes, such as learning, self-correction or reasoning [205]. Otherwise, data mining is used for knowledge discovery in data, e.g., to find new patterns and models, and is often more descriptive than predictive [204].

The book *Data Science Applied to Sustainability Analysis* [206] identifies two possible use cases for machine learning in LCI, either for data cleaning after automated data collection or for the estimation of inventory flows. Ghoroghi et al. [207] added that machine learning techniques could also be used to reduce uncertainty by increasing accuracy in LCA and reducing the simulation time. Building on their distinction, one can also divide the prediction of data into, on the one hand, prediction using similar data (such as chemicals with similar descriptors) or prediction for higher TRLs (e.g., upscaling of a lab experiment or pilot plant) on the other.

Automated data collection allows for incorporating ontology to make data for LCI more easily machine-processable and connected and can be observed, for example, in Mittal et al. [56], Lousteau-Cazalet et al. [80] and Ingwersen et al. [55]. Data (or text) mining was used and mentioned in their abstract in four literature sources [31,33,56,74]. They were all based on the need for datasets with ontologies and more linked open data.

Data uncertainty was only included by Lousteau-Cazalet et al. [80], with the pedigree matrix, though for future developments, other approaches, such as probabilistic approaches, could be possible [203].

Machine learning was only mentioned once in the title and abstract by Hou et al. [187], who evaluated different models and concluded that their random forest model was best for estimating the ecotoxicity characterization of chemicals. In addition, however, Wernet et al.,

Song et al. and Kleinekorte et al. [143,165,167] used ANNs and Gaussian process regression [170] for environmental impact estimation. None of these generate LCI data as they go directly to the LCIA step.

## 4. Discussion and Conclusions

This review describes various approaches and developments in the field of automated data generation and collection. Automation potential to support the LCA practitioners work in the life cycle inventory phase was identified in the areas of:

- Data extraction and mining (from literature, online available governmental and industry data, company documents and databases)
- Data combination and testing of compatibility (e.g., connecting back- and foreground data)
- Data insertion and optimization (process simulation software connected or combined with LCA software)
- Data generation based on limited data descriptors (process simulation, calculations and stoichiometry)
- Data generation based on data from similar in- and outputs (such as molecular modeling and proxy data)

In addition, database developers use prioritization calculations to improve the efficiency and effects of their database updates.

There could be differences in the development and further usage of the methods/models identified (see Figure 11), where the most developed and most frequently used of the 30 reviewed papers are the combination and testing of compatibility (LCI Database tools), data generation based on more limited data through process simulation and data generation based on data from similar in- and outputs (such as molecular modeling). These data generation sources were also identified as the most frequently used sources, with process simulations being most frequently used (12), followed by molecular-structure-based models (6) and LCI databases (6). Focusing on chemistry, process simulation (4) and molecular modeling (5) were most often used for chemical production and process development (4) for the special case of biorefineries. One reason for this could be the close connections between scientists dealing with LCAs in the chemical field with tools, such as process simulation and molecular modeling. It may also be due to the easier access to data than when cooperating with companies or using statistical data, which may present additional barriers, such as lack of experience or reduced availability. Proxy data as well as stoichiometry and process calculation may also not be used that often based on their high uncertainty, even though using these methods can lead to more realistic results than omitting data. On the positive side, the visualization in Figure 11 shows that there are three to six possible data sources per TRL stage, the combination of which can help generate higher-quality LCA results.

As used within other automation cases, data extraction and automated linkage could also be tested with common process simulation software to identify relevant information from the results and insert in the LCA. More automated interaction in the process simulation and LCA software could lead to

- Enhanced accuracy, as process software provides detailed information on the energy and material flows within a process
- More iterative designs, as changes in the process can be evaluated in terms of their environmental impact in real time, leading to process optimization and material efficiency improvements
- Advanced sensitivity analysis with realistic process values to identify the most significant drivers of environmental impact
- Advanced scenario analysis for changes in process parameters, energy sources or raw materials

If the integration in the interface of standardized process simulation software is limiting the use cases, this could be circumvented by using export, files e.g., in JSON. However, this procedure runs contrary to the aim of developing further to process synthesis,

where the LCA results interact with the simulation and its parameter settings (e.g., by optimization). One way of achieving this is the creation of individual process simulation software with integrated LCA modules such as BioSTEAM [78]. Its practicability has to be tested over time, as software such as this may not have the variety of application scenarios or have limited models and data to choose from. If larger process simulation software companies integrate LCAs in the future, this would support the forwarding of data and, in the best case, enable parameter interaction, leading to automatic optimization of processes according to environmental indicators. However, the usage of other data sources and functions of the LCA component included in this software could be limited. As an interface to manage interaction and data transfer, CAPE OPEN could also be an interesting and already existing interface between process simulation software. Preparatory work for this was conducted in the process sustainability prediction framework [100,101].

Molecular-structure-based models are computational models that use information about the molecular structure of a material to predict its properties and behavior. These models have the potential to improve the accuracy of Life Cycle Assessment (LCA) by providing more detailed information on the environmental impacts of materials and bear a lot of automatization potential, beyond the focus of this paper, focusing on LCI data generation and predicting environmental impacts for missing chemicals.

Other potential uses include material substitution (identifying materials with similar properties that can be used to replace materials with high environmental impacts), ecodesign (by optimizing the molecular structure of chemicals to minimize their toxicity or reduce their contribution to climate change) or improving life cycle inventory (by increasing accuracy by providing more detailed information on the chemical composition of materials and energy and resource requirements). All of those could be very beneficial when having an interface or being integrated with LCA software to automatically generate various process and input options.

However, the molecular-structure-based models reviewed in the literature of this publication have only been used for the prediction of environmental indicators, not for LCI data. This limits their applicability and topicality, as developments in the environmental assessment methods have to be incorporated manually and additional factors such as process conditions cannot be integrated in the calculation. Therefore, for further development, a combination with the similarity-based link prediction used by Hou et al. [196] and Xu et al. [197] could be beneficial, as this method, using proxy data from Ecoinvent, can deliver an average of possible process conditions and further descriptors for the chemicals. Nevertheless, the development of the molecular-structure-based models does not seem to be favored by the scientific community. The R-Package FineChem, which is also the basis for the graphical interface Estimol, was developed in 2009 and has not been updated since then, leading to software using no longer current Ecoinvent data and impact assessment methods. Although FineChem was used in later publications by co-authors of the first application until 2018, no improvements have been made since. For recent papers, creating ANNs and using Gaussian process regression, no further usage or improvements were identified either.

The review also showed that knowledge engineering, such as building and using ontologies for linked open data, is an important method of data generation. Knowledge engineering is the process of designing and developing knowledge-based systems that use artificial intelligence techniques to solve problems and make decisions. The wide range of areas of application in the scanned literature showed data extraction from text or structured data files used data sources, such as scientific literature, statistics from governmental authorities and industry associations, plant and manufacturing data as well as data from different databases. Successes for a widespread establishment of this standard can be seen, for example, through the Federal LCA Commons and should also be carried further by the Bonsai Initiative in Europe [49]. However, the flagship projects are mainly LCA internally focused on existing LCA databases, while the facilitated generation of new data through

cooperation with other data sources, such as industry, companies or scientific literature, is not yet very advanced.

Improvements in using knowledge engineering in LCA should, however, be extended, as they can provide several benefits, including improved accuracy, reductions in human error, time and cost savings, scalability potential (especially for large datasets and complex systems), standardization and the possibility of enabling the integration of LCA with other systems, such as supply chain management software or product design tools, to provide real-time feedback on the environmental impact of products and services.

Transparency is a critical aspect of Life Cycle Assessment (LCA) as it enables stakeholders to understand the assumptions, data and methods used in the assessment. Although open science is promoted by international platforms (United Nations, EU, US government), the availability of source codes is still not very high in the scanned literature, with 20% (6 out of 30) being freely available, which demonstrates a lack of traceability and reusability. Additionally, the degree of further use of the developed methods and software (at 30%) raises concerns as to how successful and easily adaptable their developments are for other scientists. Further usage of models should, in general, be supported, as shown by the fact that those few papers where the models and methods were mentioned again after publication were only those where one of the authors of the first publication was involved. However, methods should be established to raise awareness among other scientists of the applications, knowledge and motivation for use, both to ensure positive impact for LCA practice as well as making further development more likely. This aspect also includes the need for more transparency and honesty on the success and future prospects of the developed model in order to be able to map out optimal future paths by drawing on the knowledge of all the scientists involved. Another area for improvement is honesty in comparisons with other models—e.g., comparisons should not be made with the other authors' superseded older models, which leaves the impression that the model presented is deliberately being parlayed. Of advantage for further use are those that work with open-source LCA software such as Brightway [60], where the whole LCA can easily be published and recalculated from others. Reproducibility not only helps to raise the quality and trust in LCA but also simplifies an implementation of new methods by other scientists, thus making further development of the method more likely.

For use as an LCA practitioner: depending on their prior knowledge, new skills must be acquired or cooperation with other scientists must be undertaken to correctly apply certain methods, such as process simulation (e.g., BioSTEAM [78]) or machine learning techniques (e.g., using scikit-learn [190]), and to correctly assess the uncertainties. However, in the long run, these skills could be acquired when applied to several LCAs and, therefore, time resources for generating LCI data can be reduced. These tools can also increase data quality in the long term through additional data and in-depth calculation methods. Ready-made software solutions, such as FineChem [171] and Estimol ([183], an interface based on FineChem), are easier to use, but research and estimation of the actual viability of the result or a comparison with other methods are necessary to evaluate the appropriateness. Regarding software solutions based on ANNs, in addition to the limitation to the pure generation of predefined impact categories, the topicality of the data used by the ANN must also be taken into account (for those networks that are publicly accessible, the data are from 2008 and 2009). The comparisons of the different ANNs have also shown that the publication of important data, e.g., that the performance of the network varies, is vital, and attention should be paid to the underlying uncertainties when using them. However, even when referring to ready-made LCA databases, responsible research is required where the data are selected with care. As Reinhard et al. [41] showed in their data update system for Ecoinvent, the data used are still rarely checked according to system and guidelines between updates.

Generating reliable data for the different TRLs and situation fitting is and will be a challenge for the LCA community. As described, there are different solutions and methods already developed to tackle this problem but further development as well as implementa-

tions or interfaces with LCA software and the creation of appropriate framework guidelines and recommendations should be driven by international organizations such as the Life Cycle Initiative in interaction with LCA scientists and practitioners.

Focusing on chemistry, process simulation and molecular modeling, in particular, were used for chemical production and process development for the specialist field of new biorefinery concepts. As demonstrated by the results for both process simulation and molecular-structure-based models, there is still a wide range regarding how and with which methods automation is applied there, so one of the recommendations of this review is to focus on standardizing approaches and developing high-quality (preferably open-source) software solutions for both data sources. Developments by individual institutes and researchers should be made available for the public, and if methods are promising, efforts must be undertaken to develop them further. Additionally, differentiation when incorporating multiple sources would help with data quality issues as well as filling in missing data. In general, international coordination efforts should be undertaken to identify promising approaches and develop them further.

Further significant potential for automation in LCA, which could lead to more efficient and accurate assessments, should be evaluated and improved as well. Examples could be:

- Software: LCA calculations can be automated using software that can perform the required calculations based on the data collected. This saves significant time and reduces the risk of human error. Recent software developments such as the software Makersite [171] speed up the process by automatically mapping product data to multiple LCA databases.
- Sensitivity analysis: Automation can be used to perform sensitivity analysis and global sensitivity analysis on LCA models. This kind of analysis involves varying the input data and observing the impact on the overall results. Automation can speed up this process and allow for a more comprehensive analysis.
- Reporting: The results of an LCA can be presented in different formats, including tables, graphs and charts. Automation can assist in the generation of these reports, ensuring that they are standardized and consistent.
- Integration: Similar to the discussed interaction with process simulation, LCA can be integrated into other systems, such as supply chain management software, to provide real-time feedback on the environmental impact of products and services. Automation can help facilitate this integration and enable companies to make more informed decisions.

Overall, automation has the potential to streamline the LCA process, reduce the time and cost required to conduct assessments and improve the accuracy and consistency of results.

**Supplementary Materials:** The following supporting information can be downloaded at: https://www.mdpi.com/article/10.3390/su15065531/s1, Figure S1: Connection between authors in the analyzed papers summarized in Table S3. Only the analyzed papers, no further research, were used to build this figure [201]; Figure S2: Categories of the analyzed papers ($n$ = 30) by Web of Science; Figure S3: Key word analysis of 30 peer-reviewed publications used for the bibliometric and empirical evaluation; Software: VOSviewer version 1.6.18; Settings: Extraction from Title and Abstract; Minimum number of occurrences of a term—10; Selection rate = Relevance > 0.5 (non-selected terms with this condition were: LCA, life cycle assessment, framework and study); Size—Weighted after Occurrence; Scores—Average Publication year [22]; Table S1: Web of Science Analysis by Country/Region (analyzed articles see Table S3); Table S2: Web of Science Analysis by authors of the reviewed publications (source in Table S3); Table S3: Empirical Analysis of Reviewed Papers.

**Author Contributions:** Conceptualization, B.K., A.F. and B.M.-S.; methodology, data curation, formal analysis, visualization and writing—original draft preparation: B.K.; writing—review and editing, B.K., A.F., S.S.L., W.W. and B.M.-S.; supervision, A.F. and B.M.-S. All authors have read and agreed to the published version of the manuscript.

**Funding:** This research received no external funding.

**Institutional Review Board Statement:** Not applicable.

**Informed Consent Statement:** Informed consent was obtained from all subjects involved in the study.

**Data Availability Statement:** Data are contained within the article or Supplementary Material.

**Acknowledgments:** The authors would like to thank the University Library of TU Wien for the financial support through its Open Access Funding by TU Wien Program and for editing/proofreading, as well as Karl Detering for proofreading.

**Conflicts of Interest:** The authors declare no conflict of interest.

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
