# Peer review of "Automation of Life Cycle Assessment—A Critical Review of Developments in the Field of Life Cycle Inventory Analysis"

_sustainability, doi:10.3390/su15065531_

Round 1

Reviewer 1 Report

Title: Automation of Life Cycle Assessment – a critical review of developments in the field of life cycle inventory analysis

 From the perspective of data collection, this paper summarizes the current situation of automation potential of life cycle inventory phase of process engineering in detail and provides suggestions for further development of chemical and process engineering. The content is detailed and solid, and the author's work deserves recognition. However, there are still some shortcomings to be further improved. As follows:

1. The summary is verbose. The author has spent too much energy on the result display, and the logical relationship is not obvious enough, which needs further improvement.

2. The article has many mistakes in format and spelling.

4. There are duplicates in Section 2, please check carefully.

5. There are two Figure 2 in the paper. What is the manufacturing readiness levels in first Figure 2 and what is the difference from the technical readiness level?

6. The author has classified the literature using the hierarchy structure method, but the classification criteria and the specific content of the corresponding categories are not clear enough.

7. Figures in Section 3 are basically the methods in existing literature. Whether the processes of similar multiple methods can be compared to reflect the research gap, rather than being drawn directly and alone without explanation.

8. There are many subheadings in Section 3 and the format is confusing. Please check carefully or try another form.

9. The author has written the discussion and conclusion comprehensively, but the logical level can be clearer.

Author Response

Dear Sir or Madame,

Thank you for your valuable comments! I tried to incorporate all your feedback .

1) I hope the abstract is now better, as it needed a lot of reduction to reduce it to 200 words (as recommended by MDPI).

2) The paper was in proofreading, so there should not be any mistakes anymore.

4) There are no double statements in the Material & Methods section anymore- thank you for pointing that out, it must have happened when copying the text in the MDPI text format.

5) Thank you for the comment, the figure numbering should be correct now.

6) Thank you, I tried to declare in the Methods and Material Section and at the beginning of the results in more detail the structuring

7) Regarding your feedback to make similarities between the methods more visible, I included in Figure 3 a better graphical reflection on automation methods similarities in the different categories.

8) I reduced the subheadings in section 3, as you are right – the headline formats for headline 2 and 3 are very similar in the MDPI draft and therefore it is confusing on which level the subheadings are.

9) Thank you, I tried to make it more logically

Best regards

Reviewer 2 Report

In this article the authors carry out an exhaustive review of the ways that exist in the bibliography to automate the assessment of the life cycle.

They present in a correct and extensive way the evolution of the life cycle assessment (LCA). They also comment on the necessary steps to be able to carry out this evaluation.

Next, they show the objectives, scope, goals and materials used in the investigation. All well referenced and ordered.

Finally, the results are shown in a comprehensive way, well explained and with important details about it. Make the study clear. Ending with a discussion and very enlightening conclusions.

However, some aspects are missing to make the work more understandable:

- There are some references to invalid images. e.g. pages 26, 27, 32...Please review the document.

- In section 4, a well-known library in machine learning, scikit-learn, is named. A BioSTEAM platform is also named, but neither references to expand the information nor more information about them are specified. It is recommended to add some additional information in this regard, in order to carry a better reading.

Author Response

Dear Sir or Madame,

Thank you for your valuable comments! I tried to incorporate all your feedback and also had the paper proof-read by a professional.

Best regards

Reviewer 3 Report

This is a comprehensive Review for Life Cycle Assessment, however, the writing shall be improved. Most content is only a simple list for reference results, the mechanism and comments from the authors are lacking. A qualified review should summarize the references and clarify the view. Some corrections are suggested.  

The abstract is so verbose that it is difficult to capture the core idea of the article. The abstract should be an overview for the research content and important findings. The current length is not usual, and needs to be streamlined. In addition, some mportant results from references are missing.

Author Response

Dear Sir or Madame,

Thank you for your valuable comments! I tried to incorporate your feedback. I hope the abstract is now ok for you, as it needed a lot of reduction to reduce it to 200 words (as recommended by MDPI).

Best regards

Reviewer 4 Report

This study is quite interesting to me. As the author mentioned, performing a LCA requires a lot of time (especially, in data collection step). I wish the idea of the automation of LCA would reduce the discrepancy between LCA studies and lots of time to implement LCA. In terms of these thoughts, this study is important but I also recommend to modify some unclear parts.

1. Abstract is too long, please shorten this context.

2. page 5, lines 191-221 is repeated, please remove this part.

3. The Figure numbering is wrong

4. Correct all of "Error, reference source not found"

5. The font of some subheading is not consistent. It should be reorganized and unified, otherwise it will make readers confused.

Author Response

 Dear Sir or Madame,

Thank you for your valuable comments! I tried to incorporate your feedback.

I hope the abstract is now ok for you, as it needed a lot of reduction to reduce it to 200 words (as recommended by MDPI).

The formatting problems should be fine as well now and I also did professional proofreading.

Regarding the subheadings I delated all headline 3 subheadings as they looked to similar to headline 2. I hope it is better organized now.

Best regards

Reviewer 5 Report

This manuscript discussed an issue of the automation potential in the scientific literature with a focus on the life cycle inventory.  I have the following concerns:

1.      This manuscript tried to survey the current state of automation potential in the scientific literature with a focus on the life cycle inventory phase in process engineering, and showed many methods, from traditional methods to modern approaches. However, I did not see valuable conclusions in a convergent way.  The authors might need to do deeper discussion and comparison.

2.      The Abstract is too lengthy, and cannot understand the point.

3.      There are lots of statements repeated in the section of Materials and Methods.  In addition, in this section, there is no aim, goal, and method to be discussed under their titles.

4.      In the section of Results, the authors used Figure 2 to identify methods of LCI Data generation with automation potential, then discussed the methods one by one. However, I would like to suggest discussing between methods for the ease of reading.

5.      The discussion after Line 1161 in the Results section, it needs to explain why they are there with the previous discussion.

Author Response

Dear Sir or Madame,

Thank you for your valuable comments! I tried to incorporate all your feedback.

1) I went over the discussion with my colleagues and changed Fig. 3 to make the relationships more visible.

2) The abstract is shortened to around 200 words, as recommended by MDPI.

3) There are no double statements in the Material & Methods section anymore- thank you for pointing that out, it must have happened when copying the text in the MDPI text format.

4) I discussed with my co-authors your input regarding changing the way/order to discuss the results. However, we decided to leave it like this and make the structure clearer (through reducing headlines level 3). This way, to discuss methods structured by TRL, is quite typically in process engineering.

Regarding your point 5) line 1161 was in the version provided to me by MDPI, which should be the same you got, some part in the proxy method description. As this did not fit your comment, you may commented on the Bibliometric Analysis. A description of the methods is in the material and method part is included and an I went over the introduction to transition to the topic.

The paper also was in professional proofreading now, so there should not be any mistakes anymore.

Best regards

Round 2

Reviewer 5 Report

No more questions.

Author Response

Thank you for your review!